# PLMSearch: Protein language model powers accurate and fast sequence search for remote homology

Wei Liu [1], Ziye Wang [1], Ronghui You[1], Chenghan Xie[2], Hong Wei[3], Yi Xiong [4], Jianyi Yang [5] ✉ & Shanfeng Zhu [1,6,7,8,9] ✉

Homologous protein search is one of the most commonly used methods for protein annotation and analysis. Compared to structure search, detecting distant evolutionary relationships from sequences alone remains challenging. Here we propose PLMSearch (**P**rotein **L**anguage **M**odel), a homologous protein search method with only sequences as input. PLMSearch uses deep representations from a pre-trained protein language model and trains the similarity prediction model with a large number of real structure similarity. This enables PLMSearch to capture the remote homology information concealed behind the sequences. Extensive experimental results show that PLMSearch can search millions of query-target protein pairs in seconds like MMseqs2 while increasing the sensitivity by more than threefold, and is comparable to state-of-the-art structure search methods. In particular, unlike traditional sequence search methods, PLMSearch can recall most remote homology pairs with dissimilar sequences but similar structures. PLMSearch is freely available at https://dmiip.sjtu.edu.cn/PLMSearch.

Homologous protein search is a key component of bioinformatics methods used in protein function prediction[1–6], protein–protein interaction prediction[7], and protein-phenotype association prediction[8]. The goal of homologous protein search is, for each query protein, homologous proteins from the target dataset (generally a large-scale standard dataset like Swiss-Prot[9]) are needed to be found. The target protein with a higher probability of homology should be ranked higher. According to the type of input data, homologous protein search can be divided into sequence search and structure search.

Due to the low cost and large scale of sequence data, the most widely used homologous protein search methods are based on sequence similarity, such as MMseqs2[10], BLASTp[11], and Diamond[12].

Despite the success of homology inference based on sequence similarity, it remains challenging to detect distant evolutionary relationships from sequences only[13]. Sequence profiles and profile hidden Markov models (HMMs) are condensed representations of multiple sequence alignment (MSAs), which specify for each position the probability of observing each of the 20 amino acids in evolutionarily related proteins. When the sequence identity is lower than 0.3, methods based on profile HMMs such as HMMER[14], HHsearch[15], and HHblits[16,17] are better tools for homologous protein search.

In scenarios involving highly distant evolutionary relationships, sequences may have diverged to such an extent that detecting their relatedness becomes challenging. Since structures diverge much more

[1]Institute of Science and Technology for Brain-Inspired Intelligence and MOE Frontiers Center for Brain Science, Fudan University, 200433 Shanghai, China. [2]School of Mathematical Sciences, Fudan University, 200433 Shanghai, China. [3]School of Mathematical Sciences, Nankai University, 300071 Tianjin, China. [4]Department of Bioinformatics and Biostatistics, Shanghai Jiao Tong University, 200240 Shanghai, China. [5]Ministry of Education Frontiers Science Center for Nonlinear Expectations, Research Center for Mathematics and Interdisciplinary Science, Shandong University, 266237 Qingdao, China. [6]Shanghai Qi Zhi Institute, Shanghai, China. [7]Key Laboratory of Computational Neuroscience and Brain-Inspired Intelligence (Fudan University), Ministry of Education, Shanghai, China. [8]Shanghai Key Lab of Intelligent Information Processing and Shanghai Institute of Artificial Intelligence Algorithm, Fudan University, Shanghai, China. [9]Zhangjiang Fudan International Innovation Center, Shanghai, China. ✉e-mail: yangjy@sdu.edu.cn; zhusf@fudan.edu.cn

slowly than sequences, detecting similarity between protein structures by 3D superposition provides higher sensitivity[18]. Protein structure search methods can be divided into (1) contact/distance map-based, such as Map_align[19], EigenTHREADER[20], and DiscoVER[21]; (2) structural alphabet-based, such as 3D-BLAST-SW[22], CLE-SW[23], Foldseek, and Foldseek-TM[24]; (3) structural alignment-based, such as CE[25], Dali[26], and TM-align[27,28]. Protein structure prediction methods (like AlphaFold2) and AlphaFold Protein Structure Database (AFDB) have greatly reduced the cost of obtaining protein structures[29–31], which expands the usage scenarios of the structure search methods. However, in the vast majority of cases, the sequence search method is still faster and more convenient. This is notably evident in scenarios involving a large number of new sequences, such as metagenomic sequences[32], sequences generated by protein engineering[33], and antibody variant sequences[34].

At the same time, protein language models (PLMs) such as ESMs[35–37] and ProtTrans[38] only take protein sequences as input, trained on hundreds of millions of unlabeled protein sequences using self-supervised tasks such as masked amino acid prediction. PLMs perform well in various downstream tasks[39], especially in structure-related tasks like secondary structure prediction and contact prediction[40]. More recently, ProtENN[41] uses an ensemble deep learning framework that

generated protein sequence embeddings to classify protein domains into Pfam families[42]; CATHe[43] trains an ANN on embeddings from the PLM ProtT5[38] to detect remote homologs for CATH[44] superfamilies; Embedding-based annotation transfer (EAT)[45] uses Euclidean distance between vector representations (initialized from ProtT5 embeddings) of proteins to transfer annotations from a set of labeled lookup protein embeddings to query protein embeddings; DEDAL[46], DeepBLAST[47], and latest pLM-BLAST[48] obtain a continuous representation of protein sequences that, combined with the Smith-Waterman (SW)[49] or Needleman-Wunsch (NW)[50] algorithm, leads to a more accurate pairwise sequence alignment and homology detection method. These methods apply representations generated by deep learning models to protein domain classification, protein annotation, and pairwise sequence alignment, fully demonstrating the advantage of deep learning models in identifying remote homology. However, protein language models are not fully utilized for the large-scale protein sequence search.

To improve the sensitivity while maintaining the universality and efficiency of sequence search, we propose PLMSearch (Fig. 1a-c). PLMSearch mainly consists of the following three steps: (1) PfamClan filters out protein pairs that share the same Pfam clan domain[42]. (2) SS-predictor (Structural Similarity predictor) predicts the similarity

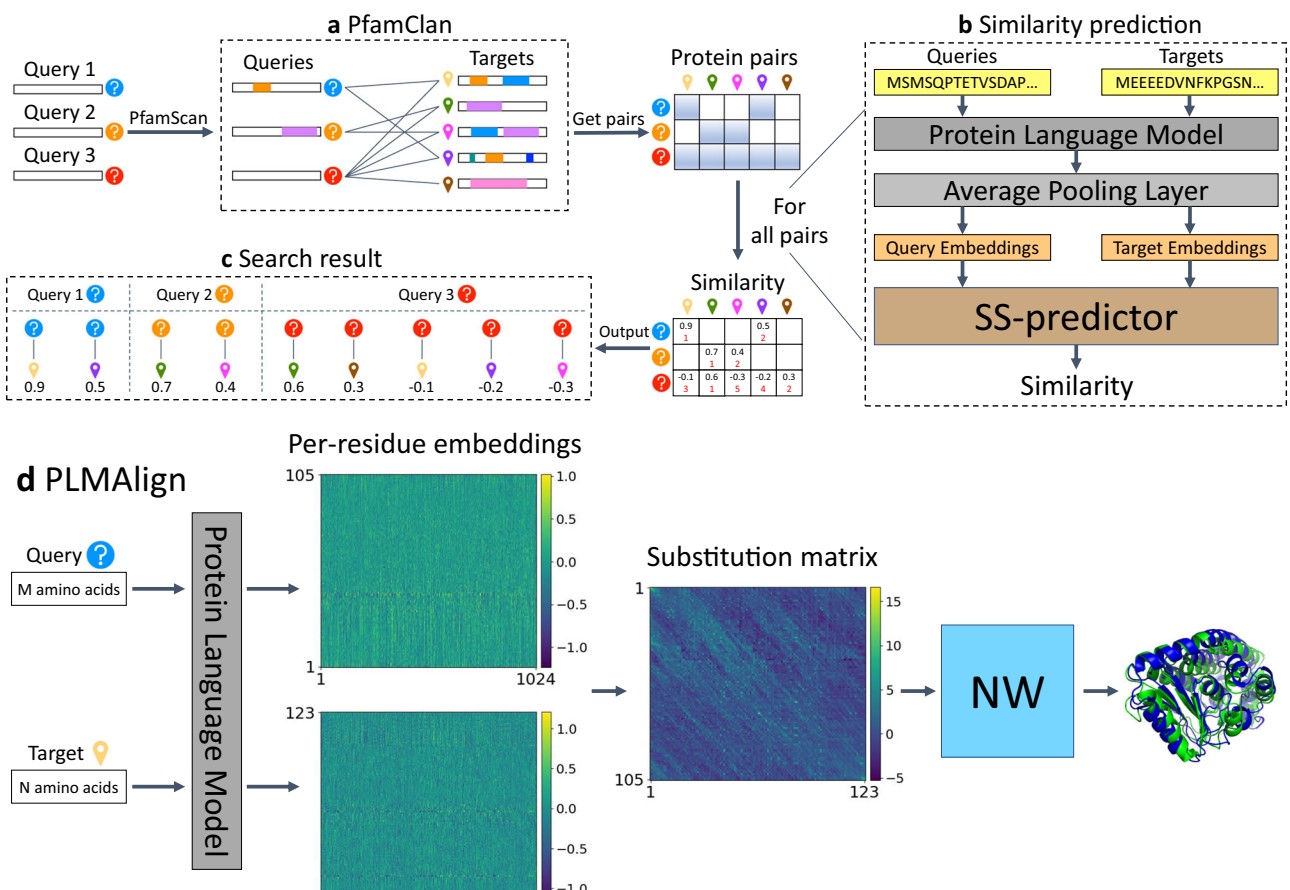

**Fig. 1 | Overview of the PLMSearch pipeline. a** PfamClan. Initially, PfamScan[54] identifies the Pfam clan domains of the query protein sequences, which are depicted in different color blocks. Subsequently, PfamClan searches the target dataset for proteins sharing the same Pfam clan domain with the query proteins. Notably, the last query protein lacks any Pfam clan domain, and therefore, its all pairs with target proteins are retained. **b** Similarity prediction. The protein language model generates deep sequence embeddings for query and target proteins. Subsequently, SS-predictor predicts the similarity of all query-target pairs. **c** Search result. Finally, PLMSearch selects the similarity of the protein pairs pre-filtered by

PfamClan, sorts these protein pairs based on their predicted similarity, and outputs the search results for each query protein separately. **d** PLMAlign. PLMAlign utilizes per-residue embeddings as input to compute a substitution matrix. This substitution matrix is then employed to replace the static substitution matrix in the Smith-Waterman (SW)[49] or Needleman-Wunsch (NW)[50] algorithm, enabling the local or global sequence alignment. The global alignment is illustrated in the figure, where the length of the query protein is 105, the length of the target protein is 123, and the embedding dimension of ProtT5-XL-UniRef50 used by PLMAlign is 1024.

between all query-target pairs with embeddings generated by the protein language model. PLMSearch will not lose much sensitivity without structures as input, because it uses the protein language model to capture remote homology information from deep sequence embeddings. In addition, the SS-predictor used in this step uses the structural similarity (TM-score) as the ground truth for training. This allows PLMSearch to acquire reliable similarity even without structures as input. (3) PLMSearch sorts the pairs pre-filtered by PfamClan based on their predicted similarity and outputs the search results for each query protein accordingly. Subsequently, PLMAlign provides sequence alignments and alignment scores for top-ranked protein pairs retrieved by PLMSearch (Fig. 1d). Search tests on SCOPe40-test and Swiss-Prot reveal that PLMSearch is always one of the best methods and provides the best tradeoff between accuracy and speed. Specifically, PLMSearch can search millions of query-target protein pairs in seconds like MMseqs2, but increases the sensitivity by more than threefold, and approaches the state-of-the-art structure search methods. The improvement in sensitivity is particularly apparent in remote homology pairs.

## Results

### PLMsearch reaches similar sensitivity as structure search methods

We benchmarked the sensitivity of SS-predictor, PLMSearch, PLMSearch + PLMAlign, five other sequence search methods (MMseqs2, Blastp, HHblits, EAT, and pLM-BLAST), four structural alphabet-based search methods (3D-BLAST-SW, CLE-SW, Foldseek, and Foldseek-TM), and three structural alignment-based search methods (CE, Dali, and TM-align). We performed search tests on SCOPe40-test and Swiss-Prot after filtering homologs from the training dataset (see "Datasets", "Metrics", and "Baselines" Sections). In the SCOPe40-test dataset (2207 proteins), we performed an all-versus-all search test. Therefore, a total of 4,870,849 query-target pairs were tested for all the methods. Figure 2a–c shows the results of the 11 most competitive methods in sensitivity and speed. Supplementary Fig. 1a–c shows the results of the other two structural alphabet-based and two structural alignment-based search methods. In the Swiss-Prot search test, we randomly selected 50 queries from

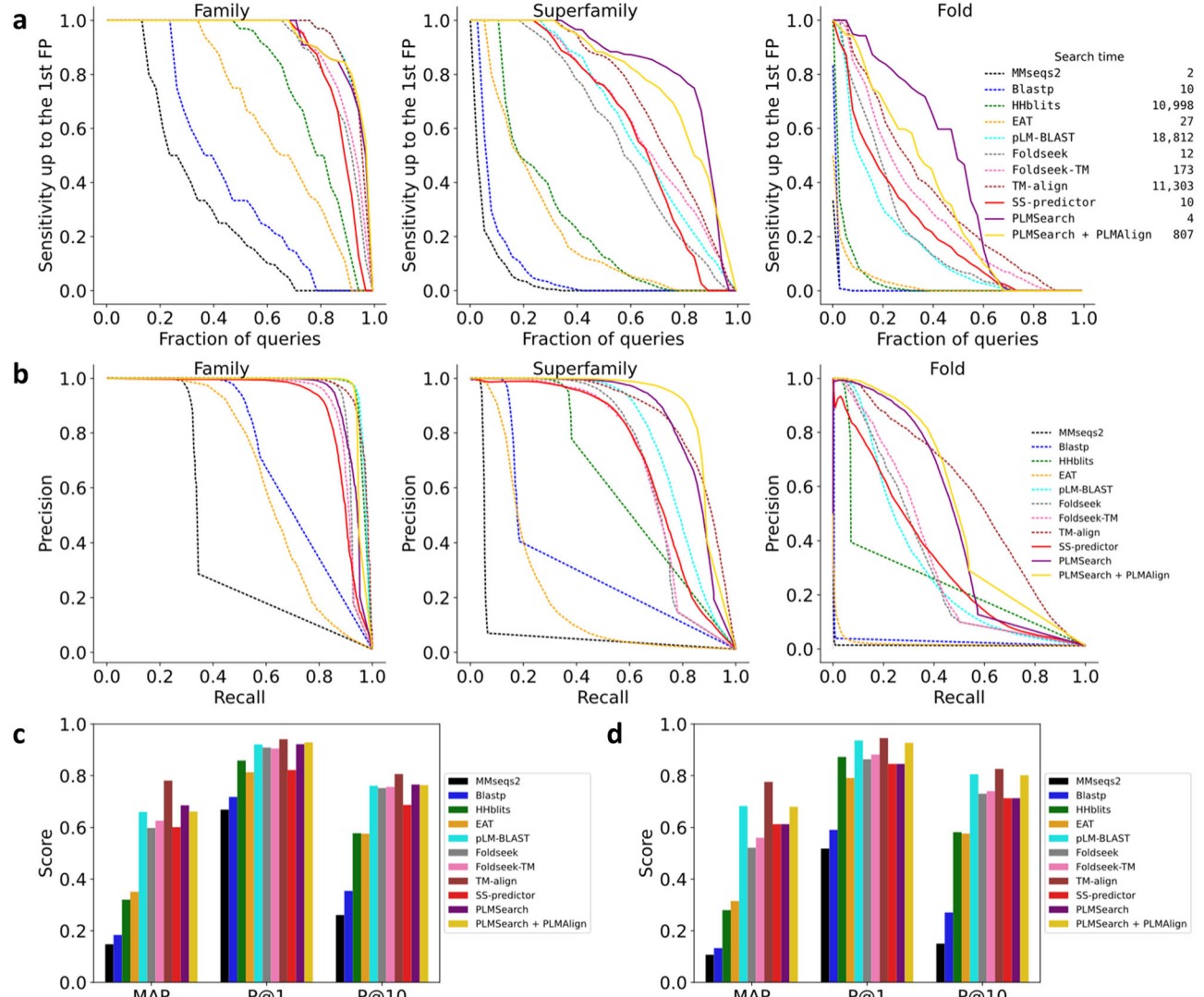

**Fig. 2 | PLMsearch reaches similar sensitivity as structure search methods.**
**a–c** The all-versus-all search test on SCOPe40-test. For family-level, superfamily-level, and fold-level recognition, TPs were defined as same family, same superfamily but different family, and same fold but different superfamily, respectively. Hits from different folds are FPs. After sorting the search result of each query according to similarity, we calculated the sensitivity as the fraction of TPs in the sorted list up to the first FP. **a** We took the mean sensitivity over all queries as AUROC. In addition,

the total search time for the all-versus-all search test with a 56-core Intel(R) Xeon(R) CPU E5-2680 v4 @ 2.40 GHz and 256 GB RAM server is shown on the legend.
**b** Precision-recall curve. **c** MAP, P@1, and P@10. **d** Evaluation on new proteins (see "New protein search test" Section). Supplementary Table 2 and Supplementary Table 4 record the specific values of each metric. Source data are provided as a Source Data file.

Swiss-Prot and 50 queries from SCOPe40-test (a total of 100 query proteins) and searched for 430,140 target proteins in Swiss-Prot. Therefore, a total of 43,014,000 query-target pairs were tested for the six most efficient methods on searching large-scale datasets (Supplementary Fig. 2, Supplementary Table 1).

Supplementary Table 2 records the specific values of each metric for the all-versus-all search test on SCOPe40-test. PLMSearch performs well on most metrics, especially at the superfamily-level and fold-level, which are shallower and have less significant similarity between protein sequences. PLMSearch is 3, 16, and 219 times exceeding MMseqs2 in AUROC of the family-level (from 0.318 to 0.928), superfamily-level (from 0.050 to 0.826), and fold-level (from 0.002 to 0.438), respectively. Supplementary Table 3 indicates that the primary reason for the improvement is that PLMSearch is more robust and makes the rank of the first FP (**F**alse **P**ositive) lower, increasing the number of total TPs (**T**rue **P**ositives) up to the first FP (38 times exceeding MMseqs2, from 2.74 to 104.78). PLMSearch + PLMAlign uses PLMAlign to align protein pairs with a similarity exceeding 0.3 from PLMSearch. The alignment score is then used to rerank, which improves the precision rate, especially on the search test for new proteins that fail to scan out any Pfam domain (Fig. 2d, Supplementary Fig. 1d, Supplementary Table 4). Accordingly, the exclusion of pairs with a similarity below 0.3 leads to a marginal decrease in the recall rate.

## PLMSearch searches millions of query-target pairs in seconds

We first compared the total search time of different methods for the all-versus-all search test on SCOPe40-test (2,207 proteins, 4,870,849 query-target pairs). To ensure fairness, we used the same computing resources (a 56-core Intel(R) Xeon(R) CPU E5-2680 v4 @ 2.40 GHz and 256 GB RAM server) when implementing different methods in the evaluation. For HHblits, pLM-BLAST, TM-align, and various other parallelizable methods, we utilized all 56 cores by default. Moreover, almost all methods need to preprocess the target dataset in advance. This part of the time is not required while searching, so we did not include the preprocessing time in the search time statistics. As shown in the legend of Fig. 2a, by using SS-predictor to predict the similarity instead of calculating the structural similarity (TM-score) of all protein pairs, SS-predictor (10 s) and PLMSearch (4 s) are among the fastest methods, and more than four orders of magnitude faster than TM-align (11,303 s).

PLMSearch can achieve similar efficiency on our publicly available web server with CPU only (64 * Intel(R) Xeon(R) CPU E5-2682 v4 @ 2.50 GHz and 512 GB RAM). Searching a query against Swiss-Prot (568K proteins) and UniRef50 (53.6M proteins)[9] and employing PLMAlign to align the query with the Top-10 targets requires approximately 0.15 min and 1.1 min, respectively (Supplementary Table 5). In fact, when searching a query against Swiss-Prot (568K proteins), PLMAlign takes up 0.12 min (more than 80% of the total time), and PLMSearch only takes about 0.03 minutes (Supplementary Table 6). This is because PLMSearch generates and preloads the embeddings of all target proteins in advance. This strategy helps to save much time by avoiding repeated forward propagations of the protein language model with a large number of parameters and saves the time for loading embeddings from disk to RAM. As a result, just one forward pass through the SS-predictor network is required to predict the similarity of millions of query-target pairs.

## PLMSearch accurately detects remote homology pairs

Remote homology pairs generally refer to homologous protein pairs with dissimilar sequences but similar structures[51]. Such protein pairs have low sequence similarity, so their homology is difficult to be detected by sequence alignment-based methods (MMseqs2, Blastp), but can be detected by structure-based search methods (Foldseek, Foldseek-TM, TM-align) (Fig. 3a). In this study, pairs with similar sequences and similar structures are defined as sequence identity > 0.3[51] and TM-score > 0.5[52,53] and are called "easy pairs"; pairs with dissimilar sequences but similar structures are defined as sequence identity < 0.3 but TM-score > 0.5 and are called "remote homology pairs". We conducted a specific analysis of recalled pairs and missed pairs (defined in Fig. 3b). We calculated the TM-score and the sequence identity (see "Sequence alignment" Supplementary Section) of the recalled/missed pairs and projected them onto a 2D scatter plot (Fig. 3c–h).

Compared with easy pairs (the first quadrant in Fig. 3c–h), remote homology pairs (the fourth quadrant in Fig. 3c–h) in the "twilight zone" of protein sequence homology are more difficult to detect[51]. Among the six methods (Supplementary Table 7), even the least sensitive methods MMseqs2 and Blastp recall all the easy pairs (574/574), but perform poorly on remote homology pairs (MMseqs2: 183/1105, Blastp: 203/1105). In contrast, powered by the protein language model, SS-predictor and PLMSearch search out most of the remote homology pairs (SS-predictor: 1022/1105, PLMSearch: 1087/1105, six times exceeding MMseqs2), and the recall rate exceeds Foldseek, which directly uses structural data as input (Foldseek: 934/1105, Foldseek-TM: 940/1105).

## Ablation experiments: PfamClan, SS-predictor, and PLMAlign make PLMSearch more robust

To evaluate PLMSearch without the PfamClan component, we screened a total of 110 queries from the 2207 queries in SCOPe40-test, which failed to scan any Pfam domain (see "New protein search test" Section). As expected, the performance of PLMSearch is exactly the same as that of SS-predictor, because PfamClan does not filter out any protein pair, whereas PLMSearch still achieves relatively sensitive search results (MAP = 0.612, P@1 = 0.845, P@10 = 0.712, see Supplementary Fig. 3e, Supplementary Table 4). Using PLMAlign to align and rank based on alignment scores significantly enhances precision. This improvement stems from the fact that, unlike SS-predictor, PLMAlign employs per-residue embeddings rather than per-protein embeddings as input and uses pairwise alignment instead of large-scale similarity prediction. Besides, it is noteworthy that both SS-predictor + PLMAlign and PLMSearch + PLMAlign only align pairs from SS-predictor and PLMSearch pre-filter results with a similarity exceeding 0.3 (totaling 1,591,492 and 379,707 pairs, respectively), in contrast to aligning all pairs like PLMAlign/pLM-BLAST (4,870,849 pairs). This streamlined approach significantly reduces the alignment time (nearly 16 times) while maintaining comparable precision, underscoring the benefits of leveraging SS-predictor and PLMSearch to pre-filter (Supplementary Fig. 3b).

To evaluate PLMSearch without the SS-predictor component, we first clustered the SCOPe40-test and Swiss-Prot datasets based on PfamClan (Supplementary Fig. 4, Supplementary Table 8). Specifically, proteins belonging to the same Pfam clan are clustered. The clustering results show a significant long-tailed distribution. After pre-filtering with PfamClan, more than 50% of the pre-filtered protein pairs (orange rectangles in the picture) come from the largest 1–2 clusters (big clusters), accounting for only a minor fraction of the overall clusters (SCOPe40-test: 0.231%; Swiss-Prot: 0.032%). Therefore, big clusters will result in a significant number of irrelevant protein pairs in the pre-filtering results, reducing accuracy, and must be further sorted and filtered based on similarity, which is what the SS-predictor does.

Furthermore, among all similarity-based search methods (See "Baselines"), we further compared the correlation between the predicted similarity and TM-score (Supplementary Fig. 3a). The correlation between the similarity predicted by Euclidean (COS) and TM-score is not high, resulting in a large number of actually dissimilar protein pairs ranking first. The similarity predicted by the SS-predictor is more correlated with TM-score (with a higher Spearman correlation coefficient). This is why SS-predictor outperforms other similarity-based search methods with the same embeddings as input (Supplementary Fig. 3b-e, Supplementary Table 2, Supplementary Table 4).

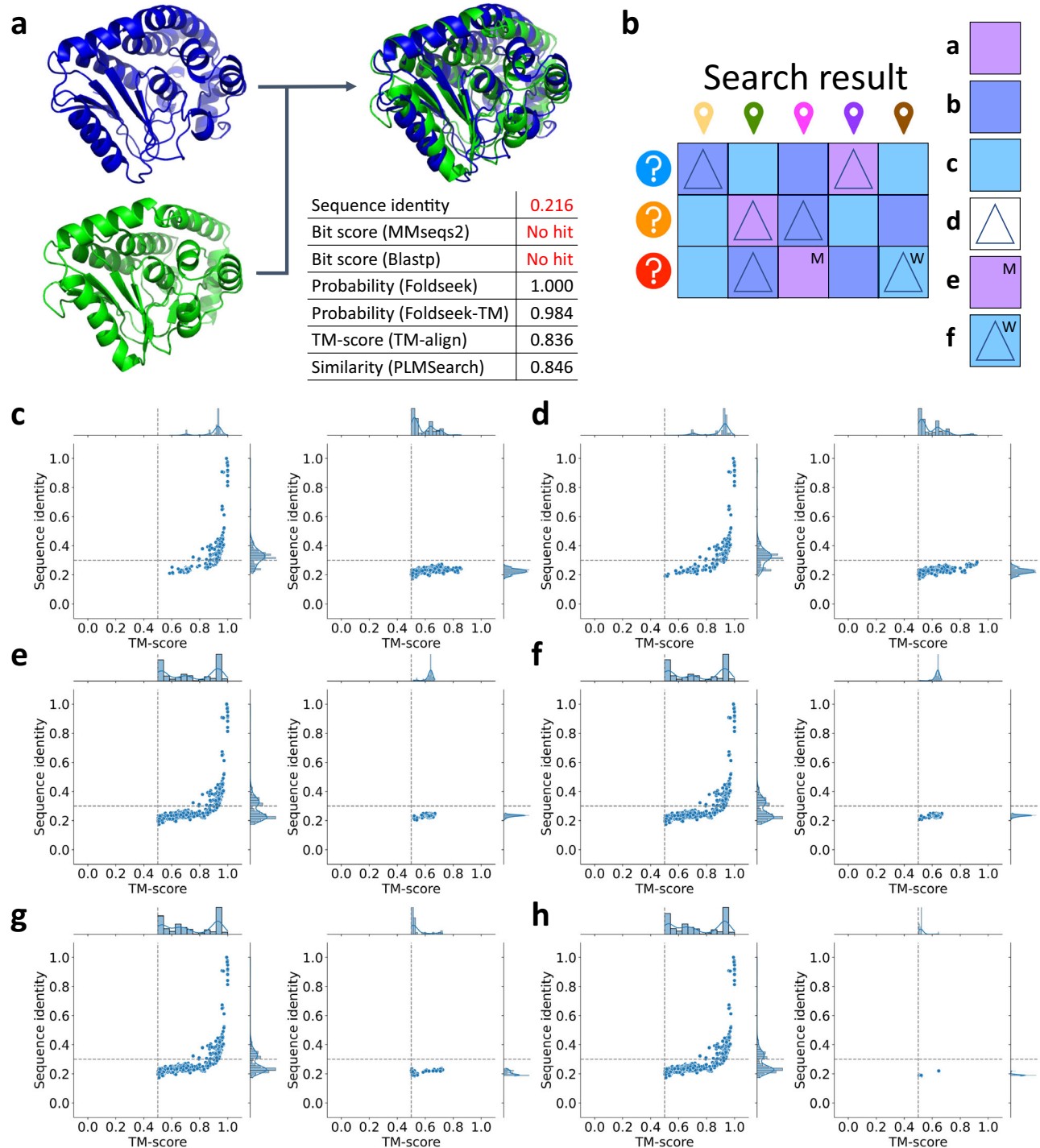

**Fig. 3 | PLMSearch accurately detects remote homology pairs. a** Case study. The sequence identity between the blue structure and the green structure is low (0.216 < 0.3) but they share similar structures. Foldseek, Foldseek-TM, TM-align, and PLMSearch recall the remote homology pair that is missed by MMseqs2 and Blastp. **b** Definition diagram. **Legend a** marks three pairs with a TM-score > 0.5, usually assumed to have the same fold[52,53]. **Legend b** marks six pairs with a TM-score between 0.2 and 0.5. **Legend c** marks six pairs with a TM-score < 0.2, usually assumed as randomly selected irrelevant pairs[52,53]. **Legend d** marks six filtered pairs. **Legend e** marks the pair at (3,3) with a TM-score > 0.5 but is not filtered out, which is a "Missed" pair. Correspondingly, protein pairs in (1,4) and (2,2) are "Recalled" pairs. **Legend f** marks the pair at (3,5) with a TM-score < 0.2 but is filtered out, which is a "Wrong" pair. **c**–**h** From the search results of five randomly selected queries to avoid oversampling (with Swiss-Prot as the target dataset, a total of 2,150,700 query-target pairs), we selected the 5000 pairs with the highest similarity for different search methods and counted the recalled and missed pairs: **c** MMseqs2. **d** Blastp. **e** Foldseek. **f** Foldseek-TM. **g** SS-predictor. **h** PLMSearch. For recalled pairs (left) and missed pairs (right) in each subplot, the TM-score (x-axis) and sequence identity (y-axis) are shown on the 2D scatter plot. The thresholds, sequence identity = 0.3[51] and TM-score = 0.5[52,53], are shown by dashed lines. All methods successfully recall the easy pairs in the first quadrant. But for remote homology pairs in the fourth quadrant, SS-predictor & PLMSearch did the best, followed by Foldseek & Foldseek-TM, and MMseqs2 & Blastp were the worst. Supplementary Table 7 records the specific values of each metric. Source data are provided as a Source Data file.

## Discussion

We study the use of protein language models for large-scale homologous protein search in this work. We propose PLMSearch, which takes only sequences as input and searches for homologous proteins using the protein language model and Pfam sequence analysis, allowing PLMSearch to extract remote homology information buried behind sequences. Subsequently, PLMAlign is used to align protein pairs retrieved by PLMSearch and obtain the alignment scores. Experiments reveal that PLMSearch outperforms MMseqs2 in terms of sensitivity and is comparable to the state-of-the-art structural search approaches. The improvement is especially noticeable in remote homology pairs. PLMSearch, on the other hand, is one of the fastest search methods in comparison to other baselines, capable of searching millions of query-target protein pairs in seconds. We also summarized the 11 most competitive approaches based on their input and performance in Supplementary Table 9.

We discuss the differences between search methods (like PLMSearch) and alignment methods (like pLM-BLAST and PLMAlign) in detail in Supplementary Table 10. It is noteworthy that residue embedding-based alignment methods, such as PLMAlign and pLM-BLAST[48], achieve respectable sensitivity. However, a primary limitation lies in the maximum size of the target dataset. This is particularly evident in two key aspects: (1) Residue embedding-based alignment necessitates retaining the embeddings of all residues for each protein in the target dataset, denoted as $N*L_i*D$ (where $N$ is the number of proteins, $L_i$ is the length of the protein, and $D$ is the embedding dimension). In contrast, PLMSearch only requires retaining per-protein embeddings, expressed as $N*1*D$. This results in a size difference exceeding three orders of magnitude, posing a significant challenge when implementing a dataset with the size of UniRef50, which contains 53.6 million proteins[9]. (2) Residue embedding-based alignment determines the similarity between protein pairs through pairwise global (local) alignments. In contrast, PLMSearch only needs a single forward pass through the SS-predictor network to predict the similarity of millions of query-target pairs. However, it is important to note that PLMSearch can solely predict the similarity of protein pairs without any alignment suggestions. To this end, PLMSearch + PLMAlign provides alignment for protein pairs filtered by PLMSearch with a similarity higher than 0.3. This approach not only compensates for PLMSearch's limitations but also avoids numerous low similarity and meaningless alignments, thereby maintaining high efficiency. In the future, we intend to explore the mutual attention between query and target per-residue embeddings to provide better global and local sequence alignment results.

In summary, we believe that PLMSearch has removed the low sensitivity limitations of sequence search methods. Since the sequence is more applicable and easier to obtain than structure, PLMSearch is expected to become a more convenient large-scale homologous protein search method.

## Methods

### PLMSearch pipeline

PLMSearch consists of three steps (Fig. 1a-c). (1) PfamClan. Initially, PfamScan[54] identifies the Pfam clan domains of the query protein sequences. Subsequently, PfamClan searches the target dataset for proteins sharing the same Pfam clan domain with the query proteins. In addition, a limited number of query proteins lack any Pfam clan domain, or their Pfam clans differ from any target protein. To prevent such queries from yielding no results, all pairs between such query proteins and target proteins will be retained. (2) Similarity prediction. The protein language model generates deep sequence embeddings for query and target proteins. Subsequently, SS-predictor predicts the similarity of all query-target pairs. (3) Search result. Finally, PLMSearch selects the similarity of the protein pairs pre-filtered by PfamClan, sorts the protein pairs based on their similarity, and outputs the search results for each query protein separately.

For the top-ranked query-target pairs, PLMAlign is used to generate local or global alignments and alignment scores. In addition, we also added parallel sequence-based Needleman-Wunsch alignment and structure-based TM-align at the end of our pipeline for users to choose.

### PfamClan

PfamClan filters out protein pairs that share the same Pfam clan domain (Fig. 1a). It is worth noting that the recall rate is more important in the initial pre-filtering. PfamClan is based on a more relaxed standard of sharing the same Pfam clan domain, instead of sharing the same Pfam family domain (what PfamFamily does). This feature allows PfamClan to outperform PfamFamily in recall rate (Fig. 4) and successfully recalls high TM-score protein pairs that PfamFamily misses (Supplementary Table 11).

### Similarity prediction

Based on the protein language model and SS-predictor, PLMSearch performs further similarity prediction based on the pre-filtering results of PfamClan (Fig. 1b). The motivation is that the clustering results based on PfamClan show a significant long-tailed distribution (Supplementary Fig. 4). As the size of the dataset increases, the number of proteins contained in big clusters will greatly expand, further leading to a rapid increase in the number of pre-filtered protein pairs

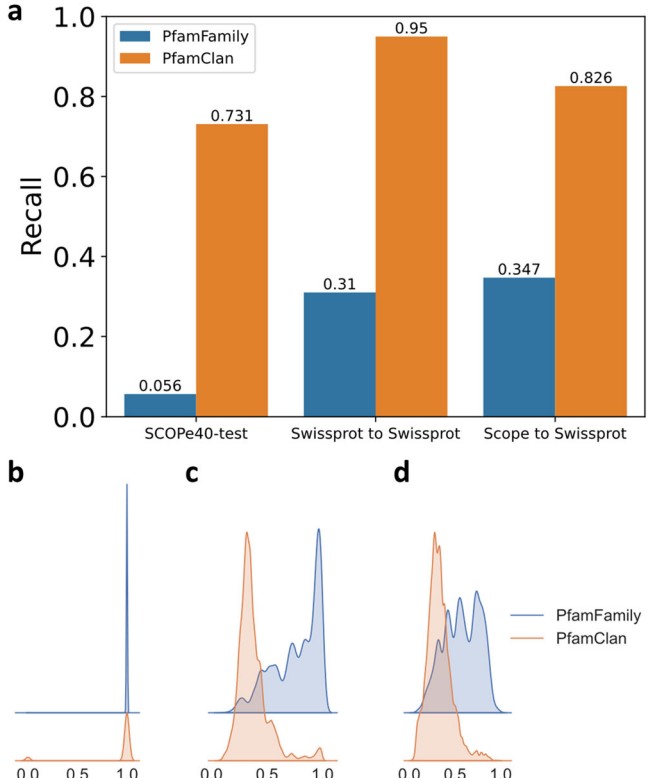

**Fig. 4 | The pre-filtering results of PfamFamily & PfamClan. a** We evaluated the pre-filtering results of PfamFamily & PfamClan on the SCOPe40-test, Swiss-Prot to Swiss-Prot, and SCOPe40-test to Swiss-Prot search tests (see "Datasets"). PfamClan achieves a higher recall rate. **b** Same(1) or Different(0) fold on SCOPe40-test. **c–d** TM-score distributions using kernel density estimation (smoothed histogram using a Gaussian kernel with the width automatically determined). **c** Swiss-Prot to Swiss-Prot; **d** SCOPe40-test to Swiss-Prot. The distribution of PfamFamily is overall to the right, because the requirements of PfamFamily are stricter than PfamClan, so the protein pair it recalls has a higher probability of being in the same fold and sharing a higher TM-score. However, this also leads to PfamFamily having a lower recall rate and missing some homologous protein pairs as shown in Supplementary Table 11. It is worth noting that the recall rate is more important in the initial pre-filtering. Source data are provided as a Source Data file.

(Supplementary Table 8). The required computing resources are excessive with TM-align used for all the filtered pairs. Instead, PLMSearch employs SS-predictor to predict similarity, which increases speed and eliminates reliance on structures.

As shown in Fig. 5a, the input protein sequences are first sent to the protein language model (ESM-1b here) to generate per-residue embeddings ($m*d$, where $m$ is the sequence length and $d$ is the dimension of the vector), and the per-protein embedding ($1*d$) is obtained through the average pooling layer. Subsequently, SS-predictor predicts the structural similarity (TM-score) between proteins through a bilinear projection network (Fig. 5b). However, it is difficult for the predicted TM-score to directly rank protein pairs with extremely high sequence identity (often differing by only a few residues). This is because SS-predictor was trained on SCOPe40-train and CATHS40, where protein pairs share sequence identity < 0.4, so cases with extremely high sequence identity were not included. At the same time, COS similarity performs well in cases with extremely high sequence identity (see P@1 in Supplementary Fig. 3d-e) but becomes increasingly insensitive to targets after Top-10. Therefore, the final similarity predicted by SS-predictor is composed of the predicted TM-scores and the COS similarity to complement each other. We studied the reference similarity of COS similarity in SCOPe40-train (Supplementary Fig. 5a-b, Supplementary Table 12). We assemble the predicted TM-score with the top COS similarity as follows. When COS similarity > 0.995, SS-predictor similarity equals COS similarity. Otherwise, SS-predictor similarity equals the predicted TM-score multiplied by COS similarity.

We also studied the reference similarity of SS-predictor similarity (Fig. 6) in SCOPe40-train. There is a clear phase transition occurring around the similarity of 0.3–0.7. Supplementary Table 12 shows that for SS-predictor, protein pairs with a similarity lower than 0.3 are usually assumed as randomly selected irrelevant protein pairs. In the ridge plot Fig. 6b, as expected, the protein pairs in the same fold and different folds are well grouped in two different similarity ranges, i.e. the protein pairs in the same fold have a higher similarity and the protein pairs in different folds have a lower one. However, since similarity and SCOP fold do not have a one-to-one correspondence, there is a small overlap. Furthermore, unlike Foldseek, which focuses on local similarity, the similarity of SS-predictor, like TM-score, focuses on global similarity (Supplementary Table 13).

## PLMAlign

For the retrieved protein pairs, PLMAlign takes per-residue embeddings as input to obtain specific alignments and alignment scores (Fig. 1d). PLMAlign subsequently uses alignment scores to rerank, which improves the ranking results further. Specifically, inspired by pLM-BLAST[48,55], PLMAlign also uses per-residue embeddings of the query-target protein pair to calculate the substitution matrix. The substitution matrix is then used in the SW/NW algorithm to perform local/global alignment. On the one hand, Compared with traditional SW/NW using a fixed substitution matrix, the substitution matrix calculated by PLMAlign uses protein embedding generated from the sequence context, thus containing deep evolutionary information. On the other hand, compared with pLM-BLAST, which uses cosine similarity, PLMAlign employs the dot product similarity and linear gap penalty. This enables PLMAlign to better align remote homology pairs while reducing the algorithm's complexity to $O(mn)$ to ensure high efficiency. The other differences between the SW/NW algorithm, pLM-BLAST, and PLMAlign are discussed in further detail in (Supplementary Table 14). Therefore, PLMAlign performs better on remote homology alignment (using "Malisam and Malidup" datasets as benchmarks, see Supplementary Fig. 6, Supplementary Table 15). Also, see "Remote homology alignment" Supplementary Section for detailed settings of PLMAlign and the evaluation of alignment results. The reference score of PLMAlign is provided in Supplementary Fig. 5c, d and Supplementary Table 12. PLMAlign in the main text uses global alignment to generate alignment scores.

## Datasets

The data volumes and uses of each dataset are summarized in Supplementary Table 16.

**SCOPe40.** The SCOPe40 dataset consists of single domains with real structures. Clustering of SCOPe 2.01[56,57] at 0.4 sequence identity yielded 11,211 non-redundant protein domain structures (SCOPe40). As done in Foldseek, domains from SCOPe40 were split 8:2 by fold into SCOPe40-train and SCOPe40-test, and then domains with a single chain were reserved. It is worth mentioning that each domain in SCOPe40-test belongs to a different fold from all domains in SCOPe40-train, so the difference between training and testing data is much larger than that of pure random division. We also studied the max sequence identity of

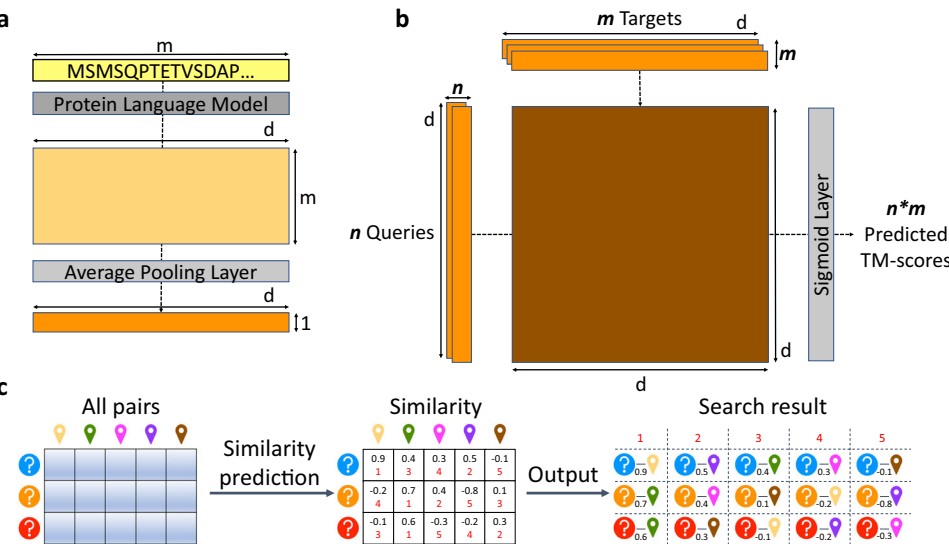

**Fig. 5 | SS-predictor. a** Protein embedding generation. The protein language model converts the protein sequence composed of $m$ residues into $m*d$ per-residue embeddings, where $d$ is the dimension of the embedding. Then the average pooling layer converts them into the corresponding $1*d$ per-protein embedding $z$. **b** SS-predictor. A bilinear projection network predicts all the TM-scores between $n$ query embeddings $z1$ and $m$ target embeddings $z2$ by $z1*W*z2$, where the matrices $W$ are the learned parameters. **c** Similarity-based search methods. The similarity of all protein pairs is predicted, sorted, and then outputted as a search result. According to different similarity, three corresponding search methods are Euclidean, COS, and SS-predictor.

**a** 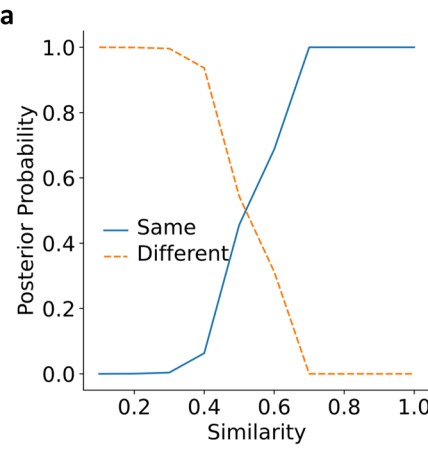

**b** 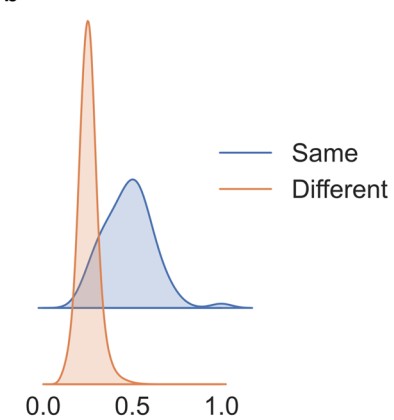

**Fig. 6 | Reference similarity. a** The posterior probability of proteins with a given similarity being in the same fold or different folds in SCOPe40-train. **b** Similarity distribution of the same and different folds protein pairs using kernel density estimation (smoothed histogram using a Gaussian kernel with the width automatically determined). The similarity of protein pairs belonging to the same protein pair is significantly higher than that of protein pairs belonging to different folds. The posterior probability corresponding to the similarity is shown in Supplementary Table 12. See "Reference similarity" Supplement Section for more details.

each protein in SCOPe40-test relative to the training dataset (Supplementary Fig. 7). From the figure, we can draw a similar conclusion that the sequences in SCOPe40-test and the training dataset are quite different, and most of the max sequence identity is between 0.2 and 0.3. PLMSearch performs well in SCOPe40-test (Fig. 2a–c, Supplementary Fig. 1a–c, Supplementary Table 2), implying that PLMSearch learns universal biological properties that are not easily captured by other sequence search methods[13].

**New protein search test.** In real-world scenarios, newly discovered and unclassified proteins often play a crucial role in innovative research. To assess the effectiveness of various methods in searching these proteins, we introduced an additional search test exclusively comprising query proteins that failed to scan any Pfam domain. Specifically, we screened a total of 110 queries from the 2207 queries in the SCOPe40-test, which failed to scan any Pfam domain. In the all-versus-all search test on the SCOPe40-test dataset, we counted the MAP, P@1, and P@10 metrics with only new proteins as queries (110 proteins) and SCOPe40-test as targets (2207 proteins).

**Swiss-Prot.** Unlike SCOPe, the Swiss-Prot dataset consists of full-length, multi-domain proteins with predicted structures, which are closer to real-world scenarios. Because the throughput of experimentally observed structures is very low and requires a lot of human and financial resources, the number of real structures in datasets like PDB[58–60] tends to be low. AlphaFold protein structure database (AFDB) obtains protein structure through deep learning prediction, so it contains the entire protein universe and gradually becomes the mainstream protein structure dataset. Therefore, in this set of tests, we used Swiss-Prot with predicted structures from AFDB as the target dataset.

Specifically, we downloaded the protein sequence from UniProt[9] and the predicted structure from the AlphaFold Protein Structure Database[31]. A total of 542,317 proteins with both sequences and predicted structures were obtained. For these proteins, we dropped low-quality proteins with an avg. pLDDT lower than 70, and left 498,659 proteins. In order to avoid possible data leakage issues, like SCOPe40, we used 0.4 sequence identity as the threshold to filter homologs in Swiss-Prot from the training dataset. Specifically, we used the previously screened 498,659 proteins as query proteins and SCOPe40-train as the target dataset. We first pre-filtered potential homologous protein pairs with MMseqs2 and calculated the sequence identity between all these pairs. The query protein from Swiss-Prot will be discarded if the sequence

identity between the query protein and any target protein is greater than or equal to 0.4. Finally, 68,519 proteins were deleted via homology filtering, leaving 430,140 proteins in Swiss-Prot, which we employed in our experiments. We also studied the max sequence identity of each protein in Swiss-Prot relative to the training dataset (Supplementary Fig. 7) and found that the vast majority of them were below 0.3.

Subsequently, we randomly selected 50 queries from Swiss-Prot and 50 queries from SCOPe40-test as query proteins (a total of 100 query proteins) and searched for 430,140 target proteins in Swiss-Prot. Therefore, a total of 43,014,000 query-target pairs were tested. The search test for 50 query proteins from Swiss-Prot and SCOPe40-test are called "Swiss-Prot to Swiss-Prot" and "SCOPe40-test to Swiss-Prot", respectively.

**CATHS40.** The SCOPe40-train dataset includes 8953 proteins and TM-scores for all protein pairs were calculated for training. As the majority of these pairs had TM-scores below 0.5, only 504,553 pairs (among 80,156,209 in total) had a TM-score above 0.5 for model training. To enhance the model's generality, we supplemented it with high-quality protein pairs extracted from the curated CATH domain dataset[44,47]. We began with the CATHS40 non-redundant dataset of protein domains, which exhibits no more than 0.4 sequence similarity. Domains exceeding 300 residues were filtered out, leaving 27,270 domains. To prevent potential data leakage issues, akin to SCOPe40, we applied a 0.4 sequence identity threshold to filter homologs in CATHS40 from the testing dataset (SCOPe40-test and Swiss-Prot). Finally, 21,474 proteins in CATHS40 were left for training, and the max sequence identity of the test dataset to the new training dataset is still less than 0.4 (Supplementary Fig. 7). We then undersampled CATHS40 domain pairs from different folds to acquire a substantial amount of training pairs with TM-scores above 0.5. Specifically, we sampled the TM-scores of 28,440,312 protein pairs for training, of which 7,813,946 pairs had a TM-score above 0.5 (Supplementary Table 16).

**Target datasets on the web server.** We currently have the following four target datasets on the web server for users to search: (1) Swiss-Port (568K proteins)[9], the original dataset without filtering; (2) PDB (680K proteins)[58–60]; (3) UniRef50 (53.6M proteins)[9]. UniRef50 is built by clustering UniRef90 seed sequences that have at least 50% sequence identity to and 80% overlap with the longest sequence in the cluster; (4) Self (the query dataset itself).

**Malisam and Malidup.** We employed two gold-standard benchmark datasets, Malisam[61] and Malidup[62], to establish a robust reference for alignment. These sets rigorously select structural alignments, emphasizing difficult-to-detect, low-sequence-identity remote homology. Malidup contains pairwise structure alignments, specifically targeting homologous domains within the same chain, thereby exemplifying structurally similar remote homologs. Malisam contains pairs of analogous motifs.

## Metrics

We evaluated different homologous protein search methods using the following four metrics: AUROC, AUPR, MAP, and P@K.

- AUROC[24]: The mean sensitivity over all queries, where the sensitivity is the fraction of TPs in the sorted list up to the first FP, all excluding self-hits.
- AUPR[24]: Area under the precision-recall curve.
- MAP[63]: Mean average precision (MAP) for a set of queries is the mean of the average precision scores for each query.

$$MAP = \frac{\sum_{q=1}^{Q} AveP(q)}{Q} \qquad (1)$$

where Q is the number of queries.

- P@K[20]: For homologous protein search, as many queries have thousands of relevant targets, and few users will be interested in getting all of them. Precision at k (P@k) is then a useful metric (e.g., P@10 corresponds to the number of relevant results among the top 10 retrieved targets). P@K here is the mean value for each query.

On SCOPe40-test, we performed an all-versus-all search test, which means both the query and the target dataset were SCOPe40-test. To make a more objective comparison, the settings used in the all-versus-all search test are exactly the same as those used in Foldseek[24]. Specifically, for family-level, superfamily-level, and fold-level recognition, TPs were defined as the same family, same superfamily but different family, and same fold but different superfamily, respectively. Hits from different folds are FPs. After sorting the search result of each query according to similarity (described in Supplementary Table 17), we calculated the sensitivity as the fraction of TPs in the sorted list up to the first FP to better reflect the requirements for low false discovery rates in automatic searches. We then took the mean sensitivity over all queries as AUROC. Additionally, we plotted weighted precision-recall curves (precision = TP/(TP+FP) and recall = TP/(TP+FN)). All counts (TP, FP, FN) were weighted by the reciprocal of their family, superfamily, or fold size. In this way, families, superfamilies, and folds contribute linearly with their size instead of quadratically[14]. MAP and P@K were calculated according to the TPs and FPs defined by fold-level.

On search tests against Swiss-Prot, without the human manual classification on SCOPe as the gold standard, proteins require a reference annotation method. Therefore, TPs were defined as pairs with a TM-score > 0.5, otherwise FPs. MAP and P@K are then calculated accordingly.

## Baselines

**Previously proposed methods.** (1) Sequence search: MMseqs2[10], BLASTp[11], HHblits[16,17], EAT[45], and pLM-BLAST[48]. (2) Structure search—structural alphabet: 3D-BLAST-SW[22], CLE-SW[23], Foldseek, and Foldseek-TM[24]. (3) Structure search—structural alignment[64]: CE[25], Dali[26], and TM-align[27,28]. For the specific introduction and settings of these proposed methods, see "Baseline details" Supplementary Section.

**Similarity-based search methods.** These methods predict and sort the similarity between all query-target pairs (Fig. 5c). Different search methods are distinguished according to the way they predict similarity.

- Euclidean: Use the reciprocal of the Euclidean distance between embeddings.

$$similarity(p,q) = \frac{1}{\sqrt{\sum_{i=1}^{n} (p_i - q_i)^2 + 1}} \qquad (2)$$

- COS: Use the COS distance between embeddings. $\epsilon$ is a small value to avoid division by zero.

$$similarity(p,q) = \frac{p \cdot q}{\max\left(\|p\|_2 \cdot \|q\|_2, \epsilon\right)} \qquad (3)$$

- SS-predictor: Combine the predicted TM-score with the top COS similarity. Unlike PLMSearch, which predicts pre-filtered pairs from PfamClan, SS-predictor extends its prediction to all protein pairs.

## Experiment settings

**Pfam result generation.** We obtained the Pfam family domains of proteins by PfamScan (version 1.6)[54] and Pfam dataset (Pfam36.0, 2023-09-12)[42]. For PfamClan, we query the comparison table Pfam-A.clans.tsv and replace the family domain with the clan domain it belongs to. For the family domain that has no corresponding clan domain, we treat it as a clan domain itself.

**Protein language model.** ESMs are a set of protein language models that have been widely used in recent years. We used ESM-1b (650M parameters)[35], a SOTA general-purpose protein language model, to efficiently generate per-protein embeddings for PLMSearch.

For PLMAlign, the more extensive ProtT5-XL-UniRef50 (3B parameters) is used to generate per-residue embeddings. We conducted a detailed evaluation and analysis of the results from ESM-1b and ProtT5-XL-UniRef50 embeddings, as elaborated in "Remote homology alignment" Supplementary Section (Supplementary Fig. 8, Supplementary Table 15). Based on the analysis, for PLMAlign, ProtT5-XL-UniRef50 was selected.

**SS-predictor training.** We used the deep learning framework PyTorch (version 1.7.1), ADAM optimizer, with MSE as the loss function to train SS-predictor. The batch size was 100, and the learning rate was 1e-6 on 200 epochs. The training ground truth was the TM-score calculated by TM-align. The datasets used for training (Supplementary Table 16) include (1) All protein pairs from SCOPe40-train (8953 proteins; 80,156,209 query-target pairs). (2) Undersampled protein pairs from CATHS40 (21,474 proteins; 28,440,312 query-target pairs). The parameters in ESM-1b were frozen and only the parameters in the bilinear projection network were trained.

**Experimental environment.** We conducted the experiments on a server with a 56-core Intel(R) Xeon(R) CPU E5-2680 v4 @ 2.40 GHz and 256 GB RAM. The environment of our publicly available web server is 64 * Intel(R) Xeon(R) CPU E5-2682 v4 @ 2.50 GHz and 512 GB RAM.

## Statistics and reproducibility

The experiments were not randomized.

## Reporting summary

Further information on research design is available in the Nature Portfolio Reporting Summary linked to this article.

## Data availability

The sequences and structures of the SCOPe40 dataset are available at https://scop.berkeley.edu. The sequences of the Swiss-Prot and UniRef50 dataset are freely available under the Creative Commons Attribution (CC BY 4.0) License at https://www.uniprot.org. The predicted structures are freely available from the AlphaFold Protein Structure Database at https://alphafold.ebi.ac.uk/download. CATH domain sequences and structures are publicly available at http://www.cathdb.info. Malidup can be found at http://prodata.swmed.edu/malidup; Malisam can be found at http://prodata.swmed.edu/malisam. Pfam is freely available under the Creative Commons Zero ('CC0') license at https://pfam.xfam.org. For PfamClan, we query the comparison table Pfam-A.clans.tsv at https://ftp.ebi.ac.uk/pub/databases/Pfam/current_release/Pfam-A.clans.tsv.gz. ESM-1b protein language model is available at https://github.com/facebookresearch/esm. ProtT5-XL-UniRef50 protein language model is available at https://github.com/agemagician/ProtTrans. The sequences of the PDB dataset are available at https://www.rcsb.org. Source data of our work is provided at: (1) PLMSearch: https://dmiip.sjtu.edu.cn/PLMSearch/static/download/plmsearch_data.tar.gz. (2) PLMAlign: https://dmiip.sjtu.edu.cn/PLMAlign/static/download/plmalign_data.tar.gz. Source data are provided with this paper.

## Code availability

PLMSearch is freely available at https://dmiip.sjtu.edu.cn/PLMSearch. PLMAlign is freely available at https://dmiip.sjtu.edu.cn/PLMAlign. PLMSearch and related tutorials are freely available to the public at GitHub https://github.com/maovshao/PLMSearch/blob/main/pipeline.ipynb. Reproducing our results and regenerating the main and Supplementary Figs. requires only one file at GitHub https://github.com/maovshao/PLMSearch/blob/main/main.ipynb. The results of PLMSearch can also be reproduced through the capsule published on Code Ocean https://doi.org/10.24433/CO.8325548.v1[65] or source code on Figshare https://doi.org/10.6084/m9.figshare.23254637[66]. Run PLMAlign and reproduce the alignment experiment in "Remote homology alignment" Supplementary Section at GitHub https://github.com/maovshao/PLMAlign, whose pipeline is modified from pLMBLAST[48,55]. pLM-BLAST is available from https://github.com/labstructbioinf/pLM-BLAST under an MIT License that allows the use, copying, modification, merging and publishing of the software, with copyright notice and permission notice included in all copies or substantial portions of the software. Structure visualizations were created in Pymol v.2.4.0 (https://github.com/schrodinger/pymol-open-source). For our PLMSearch data visualizations, we used Python version 3.8.16, Seaborn Version 0.12.2, and matplotlib-base Version 3.6.2.

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

## Acknowledgements

This work has been supported by the National Natural Science Foundation of China (Grant No. 62272105, T2225007, 62172274), the Shanghai Municipal Science and Technology Major Project (Grant No. 2018SHZDZX01), 111 Project (Grant No. B18015), the ZJ Lab, the Shanghai Research Center for Brain Science and Brain-inspired Intelligence Technology, Beijing Academy of Artificial Intelligence (BAAI) and the GHfund C (202302039294).

## Author contributions

S.Z. conceived the project. S.Z. and J.Y. supervised the project. W.L., S.Z. and J.Y. designed the research and the methodological framework. W.L. wrote the software. W.L. and Y.X. deployed web servers. H.W. sorted out the structure data. W.L., Z.W. and R.Y. analyzed the data. W.L. drafted the paper. Z.W., C.X., S.Z. and J.Y. modified the paper. All authors agree to the content of the final paper.

## Competing interests

The authors declare no competing interests.
