## [Peer Review File · Nature Communications]

PLMSearch: Protein language model powers accurate and fast sequence search for remote homologyReviewer #1 (Remarks to the Author):

This study by Liu et al. entitled "Protein language model powers accurate and fast sequence search for remote homology" presents a Jupyter Notebook named PLMSearch which wraps around existing software to implement a sequence search pipeline based on leveraging existing protein language models. The authors then selectively benchmark PLMSearch against some (but curiously not all) established and previously published sequence search and structural search software and present the claim that their software outperforms existing sequence search methods in terms of sensitivity and structure search-based methods in terms of computational speed (but not both at the same time).

Overall, this manuscript is written in a rather deceptive way, whereby the actual innovation implemented into the Jupyter Notebook is vaguely mentioned and benchmarks only selectively shown for certain combinations of competitor tools which have very well-known trade-offs between alignment speed and sensitivity. This study illustrates severe technical shortcomings, lack of innovation and fair assessment of competitor tools, and presents an insufficient experimental design. Due to the lack of sufficient experiments and fair benchmarks, I can only recommend an editorial decision to invite the authors for a rigorous major revision after which an actual assessment regarding innovation and impact of the presented work can be achieved.

Major comments:

- In general, it is not clear to me whether the authors were not aware of the existing work presented by Kaminski et al. or whether they chose on purpose not to compare their approach against this innovative proposal to combine classic sequence search with protein language models: <https://www.biorxiv.org/content/10.1101/2022.11.24.517862v2> (which was published before their work). It is paramount at this stage to compare their software against pLM-BLAST and discuss their technical and conceptual innovation and computational superiority over the pLM-BLAST approach.

- The manuscript doesn't recognise in sufficient detail that PLMSearch is a Python-Script that wraps around existing software and language models with a few internal data-manipulation steps which would usually be done as part of a typical bioinformatics analysis workflow. The authors fail to illustrate in detail what their conceptual and technical innovation is and which aspect of their analysis wasn't available to the bioinformatics community before PLMSearch existed.

- The shortcoming in explaining the innovation in this work is further magnified by the fact that benchmarking is performed and communicated highly selectively. For example, MMSeqs2 and other pairwise protein aligners were never designed and are not intended to serve for deep homology assessments, but rather focus on superior speed, because such pairwise sequence alignments can then serve as input to more sensitive approaches such as multiple sequence alignments/ Hidden Markov Models / Fold-predictors etc. However, the authors claim superior sensitivity (but vastly worse speed performance) over such pairwise approaches which is already very well known in the protein evolution and bioinformatics community and not a meaningful benchmark to begin with. In contrast, structural search comparisons against software such as TM-Align show less sensitivity, but higher computational speed (but then no comparison is shown against HHblits or HHpred or pLM-BLAST which also aim to increase speed for deep homology detection at the cost of loss in sensitivity). Overall, the entire benchmarking setup is very selective and only a comprehensive benchmark based on all existing software (HHpred, HHblits, MMSeqs2 Profile Search, Multiple Sequence Alignment based approaches, (jack)HMMer, Profile based BLAST, ESMFold, Foldseek etc) in terms of both: speed and sensitivity would be sufficient.

- The computational speed benchmarks are very limited and rather hidden in Extended Data Table A4 and in the Supplementary Information. Computational speed should be assessed against all competitor software (see previous point), for various input dataset sizes (scaling benchmark) and for parallelisation scaling (multicore CPU and or GPU if competitor software is leveraged by GPUs rather than CPUs). In addition, it is not clear how memory consumption will limit practical use-cases and what the upper-limit boundaries of the number of sequences are that can be processed in a reasonable hardware setup (personal computer/ server/ HPC).

- The methods section focuses on explaining the existing tools and approaches that are incorporated into PLMsearch rather than on the actual innovation within PLMsearch that would putatively allow this software to perform better if after presenting sufficient benchmarks PLMsearch still outperforms all competitors.

Reviewer #2 (Remarks to the Author):

Zhu et al. presented a swift fold recognition technique, PLMsearch (and SS-predictor), employing a machine learning approach that integrates the ESM-1b protein language model, released by Meta AI (Facebook). Based on the documented results, it appears that PLMsearch outperforms the pure sequence alignment-based method, MMseqs2, in terms of fold detection. However, its performance is slightly inferior to that of the structure-based alignment tool, TM-align. Although the search speed of PLMsearch surpasses TM-align, it remains slower than the majority of state-of-the-art methods, such as Foldseek. Overall, PLMsearch seems to hold a respectable position among these methods in terms of accuracy or running speed. The concept of the SS-predictor is similar to a previous work, "A novel sequence alignment algorithm based on deep learning of the protein folding code, *Bioinformatics* (2021)," but distinguishes itself by incorporating the cutting-edge protein language model into their deep learning network, which serves as a notable highlight. The deep learning-based alignment problem for sequences or sequence-structure alignment is indeed a captivating subject. Nevertheless, there exist several significant issues within this paper. In my opinion, the most crucial concern lies in the fact that the method solely reports whether two sequences (query and target) share the same fold, relying solely on a predicted TM-score, without providing any alignments. This, I believe, prevents the method from becoming a valuable tool for the scientific community. Additionally, some experimental test designs or quality assessments employed in the study are either uncommon or suffer from certain deficiencies. Please see my point-to-point comments. I hope those things will help the authors improve the quality of their work and manuscript.

Major:

1. As previously mentioned in the general comments, it is worth noting that PLMsearch (and SS-predictor) only provides information regarding whether two sequences (query and target) are the same fold with a predicted TM-score. However, it fails to offer guidance on how these sequences should be aligned, which is provided by the Foldseek method. In protein structure prediction (such as threading) or sequence-sequence alignment (can be extended to encompass multiple sequence alignment generation problems), the primary focus is always reporting the sequence-sequence alignment or sequence-structure alignment. To illustrate this point, let's consider an example: suppose the query corresponds to a single-domain protein that is similar to domain one of a three-domain protein target. Merely stating that the query and target share the same fold is of limited usefulness unless the method can align the query to domain one of the target and explicitly output this alignment for the user. Consequently, I highly recommend that the authors reconstruct their pipeline to include alignment as one of the output components.
 2. In the "2.1 PLMsearch reaches similar..." section, during the evaluation of the PLMsearch method on the SCOPe dataset and database, the authors made comparisons with TM-align, Dali, Foldseek, and MMseqs2. However, given that this evaluation methodology resembles template detection, I would suggest that the authors also compare their method with fold recognition or threading approaches, such as HHsearch. Since PLMsearch utilizes ESM-1b, which likely incorporates predicted 'contact/distance' information in its hidden states/layers, it would be better to also include comparisons with contact/distance-based fold recognition methods like Map_align, EigenThreader, DiscoVER, etc.
 3. In the "2.1 PLMsearch reaches similar..." section and throughout the SCOPe-based results section, the authors report the "total TPs up to the first FP" and emphasize their method could enrich more true positives in the ranking list. However, these assessments may not be particularly informative, as users without prior experience often focus solely on the first hit and tend to place trust in the result with the highest confidence score. Therefore, I suggest reporting the accuracy of the first hit, and you can refer to how this paper did such an analysis with SCOPe.
- EigenThreader: <https://academic.oup.com/bioinformatics/article/33/17/2684/3611272>

4. In the "2.1 PLMsearch reaches similar..." section, as well as in the method section "4.5.2 Evaluation based on the TM-score benchmark," and other related sections discussing the TM-score benchmarking, the authors employed a normalized TM-score using the average length of the query and target structures (i.e., $TM-align(\text{avg. length})$). However, this approach is not appropriate as the problem is query-based. During the TM-score benchmarking, the structure of the query should be known, and thus the TM-score should be normalized with the length of the query rather than the average length.
5. One issue I have identified in this paper is the excessive number of tests designed within the results section. This abundance of tests distracts the readers from focusing on the central topic. I recommend that the authors adopt a more streamlined approach by dedicating one experimental test to each section. Additionally, it would be beneficial to explain the underlying logic behind why PLMsearch is faster or superior, while providing relevant case studies. Currently, each section introduces approximately two tests (some even more), along with new concepts. This approach tends to divert the readers' attention towards understanding these concepts rather than evaluating the performance of PLMsearch.
6. Section 2.5 could potentially be divided into separate sections to improve clarity. The first paragraph could be relocated to other relevant results sections where it directly pertains to the specific findings. Similarly, the second paragraph could be moved to the method section, as it seems to primarily explain the behavior of PLMsearch rather than presenting extensive result-related information. By reorganizing the content in this manner, the paper's structure and flow will be enhanced, allowing for a more coherent presentation of the results and methodological details.
7. Figure S3 and the histograms throughout the paper appear to lack a clear design and can be confusing for readers. I suggest either presenting different histograms in separate figures or aligning the x-axis of the histograms together for better visual clarity and comparability. This will facilitate easier interpretation and understanding of the data presented in the histograms.
8. In the "4.4.2 Swiss-Prot" section, line 498, 5 proteins are too few to test!
9. Still, in the "4.4.2 Swiss-Prot" section, it is crucial to report the redundancy of the five Swiss-Prot proteins with the SCOPe training dataset. The authors should employ non-redundant Swiss-Prot sequences when testing their method against the SCOPe40-train dataset.
10. What is "->" in Figure 5? Please explain more about this figure, as its current representation is causing some confusion.
11. All pipeline and explanation figures in this paper utilize a DNA logo instead of a protein sequence logo. It is recommended to replace the DNA logo with a protein sequence logo to align with the focus of the paper.

Minor:

1. Line 17 on page 2, it should be "HH-suite" instead of "Hh-suite"
2. Line 18-21 on page 2, the sentence "These methods are based on the idea that homologous protein pairs with high sequence similarity often share similar molecular and cellular functions and structures." does not make sense here. It seems that the authors intended to summarize the methodology of sequence alignment methods such as BLAST and MMseqs2. However, the sentence does not effectively convey that meaning. Please revise it.

Reviewer #3 (Remarks to the Author):

Liu et al. present PLMSearch, which allows fast searching of homologous protein sequences without structure. In the data rich era of biology this is an important problem to solve. This paper could be of general interest if issues surrounding data leakage, a common problem in this field, can be addressed.

Major points

1. The strategy of randomly splitting SCOPe40 8:2 like in Foldseek is likely not sufficient to avoid data leakage for models that are directly training on the structural similarity. It means that sequences with 39% sequence identity, indicating near-certain homology, could be in both the training and the test set, which will give inflated results compared to unseen sequences. This can be studied by plotting the distribution of "closest sequence ID to the training set" for the test set

sequences. If this indicates significant overlap of the sets, it may be necessary to exclude some sequences from testing or retrain with a more stringent training set.

2. For the Swiss-Prot dataset, it appears that no filtering of homologs from the training set was done at all. This explains the high performance compared to Foldseek - there is likely significant data leakage. This can be studied using the same approach as above. Limiting the Swiss-Prot set to those with less than 30% sequence ID to sequences used for training may be sufficient here.

3. PLMSearch has considerably better performance than SS-predictor. This is a problem when extrapolating the performance of the model to new, unclassified sequences that are not in Pfam; they will presumably not find any Pfam matches and the performance will be equivalent to SS-predictor. These new cases are the ones of most interest when looking for remote homology, since anything with Pfam matches already has its homologs identified. This effect is hard to benchmark since the SCOP test set tends to be well-annotated in Pfam. However it should be discussed thoroughly.

4. It would be good to mention and compare to EAT (<https://github.com/Rostlab/EAT>) which is also a sequence-based search tool that uses contrastive learning based on structural classification. It is difficult to benchmark other trained approaches without data leakage though.

5. Is it possible to run an ablation using only the PfamClan step, i.e. ranking by the PfamClan score and not re-ranking by SS-predictor? This would give information on how much SS-predictor contributes to the performance of PLMSearch over a simple HMM search.

Minor points

1. Mention in the intro the reason why searching for homology with structures is effective, i.e. that structure is more conserved than sequence. Also describe briefly what protein language models are, their output ($n \times d$) and how they are trained.

2. Unless I missed it, the origin of the name SS-predictor is not given. It could be given when the name is first introduced to avoid confusion with Secondary Structure.

3. Figure 3 might be clearer with different colours for recalled and missed pairs.

4. In Figure 4 the E-value for MMseqs2 is given as "None" but this is ambiguous with a low E-value. Better to replace it with "No hit" or similar.

5. It would be useful to scale the run time up to larger databases, for example estimating the run time for searching UniRef90 and the AlphaFold database.

Response Letter

Dear Editors and Reviewers,

We greatly appreciate your time and efforts in reviewing the original manuscript. We thank the reviewers for the highly pertinent and helpful comments. In the revision, we have conducted additional experiments and modified our manuscript accordingly by carefully considering the reviewers' suggestions and comments.

Here we summarize the major changes made in the revised version of our manuscript.

1 Achieve a more comprehensive comparison by adding baselines and optimizing experiment settings

- Based on the opinions of the three reviewers, we have added the most advanced sequence search methods such as Blastp, HHblits, EAT, pLM-BLAST, and structure search method Foldseek-TM (more sensitive than Foldseek) as baselines to achieve a more comprehensive comparison.
- Similar to SCOPe40, we have removed redundancy on the Swiss-Prot dataset with a threshold of 0.4 sequence identity. As a result, the proteins from the Swiss-Prot dataset in the experiment, whether as query or target, have been filtered of homologs from the training set (SCOPe40-train) to prevent potential data leakage risks.
- Expand the size of the test set. Now the search test for Swiss-Prot contains a total of 100 query proteins (50 queries from the filtered Swiss-Prot and 50 queries from SCOPe40-test, respectively) and searches for the filtered Swiss-Prot (including 430,140 target proteins). Therefore, a total of 43,014,000 query-target pairs were tested.
- According to Reviewer 2's opinion, we have deleted the confusing metrics in the original ablation experiment and used the same metrics to evaluate the baselines and ablation methods. In addition, we have added metrics such as P@1 according to the comments. In general, our current evaluation metrics now mainly include:
 - In the all-versus-all search-test of SCOPe40-test, TPs are defined as same family, same superfamily but different family, and same fold but different superfamily, respectively. Hits from different folds are FPs. Evaluation metrics include AUROC, AUPR (the definition and settings are the same as Foldseek's evaluation), MAP, P@1, and P@10 (commonly used in protein homologous search, such as EigenThreader's evaluation).
 - In the search test against Swiss-Prot. Protein pairs with TM-score>0.5 are TPs, otherwise FPs. Evaluation metrics include MAP, P@1, and P@10.
- According to the comments of Reviewer 3, we have added an additional set of experiments to evaluate the search results of proteins that failed to scan any Pfam domain.

2 Add alignment results to the web server and pipeline

According to the opinion of Reviewer 2, we have added the detailed sequence alignment in our web server for the reference of users. We also added parallel sequence-based alignment and structure-based TM-align at the end of our pipeline, so that users can easily generate alignments they are interested in, and obtain the corresponding sequence identity and TM-score. The specific methods have been updated in the paper, Github, and the web server.

3 Modify the pipeline of SS-predictor to make it faster

The original pipeline of SS-predictor is to first assemble all query-target pairs and calculate the similarity. However, the process of assembling and binning would consume a lot of time, and the efficiency of calculating the similarity pair by pair was too low. Now, protein similarity is calculated through network forward propagation and the similarity between queries and all targets is calculated at one time. The network forward propagation (based on matrix multiplication between Pytorch tensors) is very fast, regardless of whether CPU or GPU is used. This significantly improves the speed of SS-predictor (more than 10 times).

4 Add UniRef50 as the target dataset to the web server

To expand the applicable scenarios and influence of PLMSearch, we have added UniRef50 (including 53,625,855 proteins, which is nearly 100 times that of Swiss-Prot) as one of the target datasets on the web server. At the same time, we have evaluated the average time required for a single query in Table R1 (Extended Data Table A4). In this revision, we have added the brief descriptions into the section of "Target datasets on web server" as follows.

"We currently have the following four target datasets for users to search: (1) Swiss-Prot (568K proteins) [1], the original dataset without de-redundancy; (2) PDB (680K proteins) [2--4]; (3) UniRef50 (53.6M proteins) [1]. UniRef50 is built by clustering UniRef90 seed sequences that have at least 50% sequence identity to and 80% overlap with the longest sequence in the cluster; (4) Self (the query dataset itself). Searching a query on Swiss-Prot and UniRef50 takes approximately 0.03 min and 0.5 min respectively (Table R1)."

Query num	1	10	100	1,000
Swiss-Prot (568K proteins)				
SS-predictor	0.2 min	0.5 min	3.9 min	32.3 min
PLMSearch	0.2 min	0.6 min	3.9 min	31.5 min
UniRef50 (53.6M proteins)				
SS-predictor	1.8 min	6.1 min	54.8 min	523.5 min
PLMSearch	1.8 min	6.3 min	54.7 min	516.4 min

Table R1 Total running time of the web server. The environment of the web server is CPU ONLY, with 64 * Intel(R) Xeon(R) CPU E5-2682 v4 @ 2.50 GHz and 512 GB RAM. The time required to search 1, 10, 100, and 1,000 query proteins with Swiss-Prot (568K proteins, the original dataset without de-redundancy) and UniRef50 (53.6M proteins) as the target dataset were counted respectively. Searching a query against Swiss-Prot and UniRef50 takes approximately 0.03 min and 0.5 min respectively.

5 Reconstruct the layout of figures and tables

We have reorganized the figures and tables of the paper in response to the reviewers' concerns and modified some expressions of the paper.

In the following, we present one-by-one responses to the editor's and reviewers' comments.

Reviewer 1

Q 1.1 *This study by Liu et al. entitled “Protein language model powers accurate and fast sequence search for remote homology” presents a Jupyter Notebook named PLMSearch which wraps around existing software to implement a sequence search pipeline based on leveraging existing protein language models. The authors then selectively benchmark PLMSearch against some (but curiously not all) established and previously published sequence search and structural search software and present the claim that their software outperforms existing sequence search methods in terms of sensitivity and structure search-based methods in terms of computational speed (but not both at the same time).*

Overall, this manuscript is written in a rather deceptive way, whereby the actual innovation implemented into the Jupyter Notebook is vaguely mentioned and benchmarks only selectively shown for certain combinations of competitor tools which have very well-known trade-offs between alignment speed and sensitivity. This study illustrates severe technical shortcomings, lack of innovation and fair assessment of competitor tools, and presents an insufficient experimental design. Due to the lack of sufficient experiments and fair benchmarks, I can only recommend an editorial decision to invite the authors for a rigorous major revision after which an actual assessment regarding innovation and impact of the presented work can be achieved.

Reply: Thank you for the comments. In our original manuscript, our baseline selection mainly referred to the latest SOTA structure search method Foldseek (2023, Nature Biotechnology, <https://www.nature.com/articles/s41587-023-01773-0>). Generally speaking, in terms of sensitivity, structure search tends to have an advantage over sequence search. Therefore, our baseline selection is mainly inclined to structure search methods with higher sensitivity. In the latest revision, for a comprehensive comparison, we have added some cutting-edge sequence search methods according to reviewers’ suggestions, including Blastp, HHblits, EAT, and pLM-BLAST. In addition, we have also added Foldseek-TM, which is more sensitive than Foldseek.

In terms of technological innovation, PfamScan has been a well-known approach for identifying domains in sequences. Protein language models, like ESM-1b, have also been widely employed in tasks including protein structure and function prediction, protein-protein interaction prediction, and protein-phenotype association prediction. However, they are not the same as our essential components, PfamClan and SS-predictor. PfamScan and ESM-1b are simply the foundation for PfamClan and SS-predictor (specifically, PfamScan and ESM-1b are used to preprocess protein sequences as the input to PfamClan and SS-predictor). Furthermore, PLMSearch distinguishes itself by incorporating the cutting-edge protein language model to predict similarity, thereby allowing fast searching of homologous protein sequences without structure, which is extremely valuable for protein sequence research in high-throughput circumstances. Finally, compared with other methods that also use protein language models (such as EAT and pLM-BLAST), PLMSearch also has its unique features. Compared with EAT which directly uses the Euclidean distance between embeddings, PLMSearch uses a neural network to predict the similarity and uses the structural similarity TM-score as the ground truth, so the search results are more sensitive. Compared with pLM-BLAST, PLMSearch uses per-protein embedding instead of per-residue embedding for similarity prediction, which greatly reduces the deployment space required for the target dataset while increasing the search speed by more than four orders of magnitude. Detailed differences between PLMSearch and other methods can be examined in Table R2 (Supplementary Table 4).

Based on the comments of the reviewers, we have made a rigorous major revision to our work, including comprehensively revising our experimental design, expanding the scale of the experiment, adding a variety of the latest sequence search methods as baselines, etc.

Q 1.2 *Major comments:*

- In general, it is not clear to me whether the authors were not aware of the existing work presented by Kaminski et al. or whether they chose on purpose not to compare their approach against this innovative proposal to combine classic sequence search with protein language models:

Methods	Input	Sensitivity	Speed	Query mode
Sequence search				
MMseqs2	Sequence	Low	Very Fast	Multi query
Blastp	Sequence	Low	Very Fast	Multi query
HHblits	Profile HMMs	High	Slow	Single query
EAT	Per-protein embedding	Low	Fast	Multi query
pLM-BLAST	Per-residue embedding	Very High	Slow	Single query
Structure search — structural alphabet				
Foldseek	Structure	High	Very Fast	Multi query
Foldseek-TM	Structure	Very High	Fast	Multi query
Structure search — structural alignment				
TM-align	Structure	Very High	Slow	Pairwise
Our methods				
SS-predictor	Per-protein embedding	High	Very Fast	Multi query
PLMSearch	Per-protein embedding	Very High	Very Fast	Multi query

Table R2 Summary of the characteristics of baselines. According to the performance on the all-versus-all search test on SCOPe40-test, the baselines are summarized according to their input, sensitivity, speed, and query mode.

<https://www.biorxiv.org/content/10.1101/2022.11.24.517862v2> (which was published before their work). It is paramount at this stage to compare their software against pLM-BLAST and discuss their technical and conceptual innovation and computational superiority over the pLM-BLAST approach.

Reply: Thank you for the comments. To be frank, we didn't notice the work of pLM-BLAST until we received the reviewer's comments. We have added pLM-BLAST as a baseline in the revised manuscript. Please see Section 2.1, Fig. R1 (Main text Fig. 2), Table R3 (Extended Data Table A1) for details.

Methods	AUROC			AUPR			MAP	P@K	
	Family	Superfamily	Fold	Family	Superfamily	Fold	MAP	P@1	P@10
Sequence search									
MMseqs2	0.318	0.050	0.002	0.430	0.091	0.014	0.147	0.668	0.260
Blastp	0.527	0.161	0.004	0.717	0.342	0.029	0.183	0.717	0.354
HHblits	0.920	0.363	0.064	0.969	0.623	0.256	0.320	0.858	0.577
EAT	0.648	0.230	0.025	0.646	0.225	0.020	0.350	0.813	0.575
pLM-BLAST	0.940	0.642	0.176	0.973	0.779	0.305	0.659	0.921	0.760
Structure search — structural alphabet									
3D-BLAST-SW	0.653	0.255	0.045	0.621	0.264	0.047	0.446	0.825	0.604
CLE-SW	0.672	0.265	0.033	0.432	0.171	0.035	0.440	0.814	0.592
Foldseek	0.883	0.584	0.214	0.921	0.703	0.320	0.598	0.908	0.751
Foldseek-TM	0.898	0.664	0.296	0.906	0.695	0.337	0.626	0.905	0.756
Structure search — structural alignment									
CE	0.847	0.527	0.148	0.882	0.627	0.245	0.618	0.897	0.734
Dali	0.923	0.702	0.281	0.948	0.814	0.454	0.702	0.927	0.790
TM-align	0.935	0.721	0.346	0.971	0.866	0.569	0.781	0.941	0.806
Our methods									
Euclidean	0.699	0.309	0.039	0.456	0.107	0.016	0.365	0.829	0.603
COS	0.705	0.316	0.040	0.514	0.130	0.017	0.367	0.830	0.606
SS-predictor (w/o COS)	0.874	0.515	0.108	0.901	0.601	0.179	0.559	0.840	0.711
SS-predictor	0.889	0.542	0.113	0.913	0.613	0.183	0.569	0.850	0.723
PLMSearch	0.917	0.733	0.298	0.925	0.768	0.312	0.683	0.926	0.770

Table R3 All-versus-all search test on the SCOPe40-test dataset of single-domain structures. The definition of AUROC, AUPR, MAP, and P@K is detailed in “Metrics” Section. The highest value achieved for each metric is highlighted in bold.

As far as pLM-BLAST (<https://www.biorxiv.org/content/10.1101/2022.11.24.517862>, 2023.7.30, Version 3) is concerned, it achieves respectable sensitivity. This is mainly because pLM-BLAST extends the concept of BLAST by replacing invariant substitution matrices with per-residue similarities between protein embeddings. Consequently, the similarity between a given pair of residues is entirely context-dependent. Such sequence context information has been shown to significantly improve the sensitivity of

6 *Response Letter*
Fig. R1 PLMsearch reaches similar sensitivities as structure search methods. **a-c**, The all-versus-all search test on SCOPe40-test. For family-level, superfamily-level, and fold-level recognition, TPs were defined as same family, same superfamily but different family, and same fold but different superfamily, respectively. Hits from different folds are FPs. After sorting the alignment result of each query according to similarity, we calculated the sensitivity as the fraction of TPs in the sorted list up to the first FP to better reflect the requirements for low false discovery rates in automatic searches. **a**, We took the mean sensitivity over all queries as AUROC. In addition, the total search time for the all-versus-all search test with a 56-core Intel(R) Xeon(R) CPU E5-2680 v4 @ 2.40 GHz and 256 GB RAM server is shown on the legend. **b**, Precision-recall curve. **c**, MAP, P@1, and P@10. **d**, MAP, P@1, and P@10 on a search test with Swiss-Prot as the target dataset. Table R3 records the specific values of each metric.

sequence search methods and enable the detection of non-trivial conserved patterns. However, as described in the original paper of pLM-BLAST, the primary limitation of pLM-BLAST is the maximum size of the target dataset (as searching enormous, redundant datasets containing millions of sequences would be computationally costly). This is mainly reflected in the following two aspects: (1) pLM-BLAST needs to use per-residue embeddings for global and local alignments, which means that it needs to retain the embeddings of all amino acids for each protein in the target dataset ($N * L_i * D$, N represents the number of proteins, L_i represents the length of the protein, and D represents the embedding dimension). In contrast, PLMsearch only needs to retain per-protein embeddings as the target dataset ($N * 1 * D$). This results in a size difference of 2-3 orders of magnitude and a significant challenge when implementing a

dataset with the size of UniRef50 containing 53.6M proteins [1]. (2) pLM-BLAST needs to calculate the similarity between protein pairs through specific global and local alignments, based on COS similarity pre-filtering but still needs 1-3 minutes for each query protein to search (according to its original text). In the running efficiency comparison done by us (the legend in Fig. R1a), pLM-BLAST searches slightly faster than the original paper (because the search size is smaller and we use all 56 cores for parallel computing), but still slower than most search methods. Overall, pLM-BLAST achieves respectable search sensitivity but poor speed at large-scale searches. However, when researchers focus on the similarity of individual protein pairs and specific precise sequence alignments, pLM-BLAST can help researchers obtain high-quality alignments based only on sequences. In this revision, we have added the brief descriptions into the section of "Discussion" as follows.

"It is worth mentioning that pLM-BLAST [5] achieves respectable sensitivity. This is mainly because pLM-BLAST extends the concept of BLAST by replacing invariant substitution matrices with per-residue similarities between protein embeddings. However, as described in the original paper of pLM-BLAST, the primary limitation of pLM-BLAST is the maximum size of the target dataset (as searching enormous, redundant datasets containing millions of sequences would be computationally costly). This is mainly reflected in the following two aspects: (1) pLM-BLAST needs to use per-residue embeddings for global and local alignments, which means that it needs to retain the embeddings of all amino acids for each protein in the target dataset ($N * L_i * D$, N represents the number of proteins, L_i represents the length of the protein, and D represents the embedding dimension). In contrast, PLMSearch only needs to retain per-protein embeddings as the target dataset ($N * 1 * D$). This results in a size difference of 2-3 orders of magnitude and a significant challenge when implementing a dataset with the size of UniRef50 containing 53.6M proteins [1]. (2) pLM-BLAST needs to calculate the similarity between protein pairs through specific global and local alignments, based on COS similarity pre-filtering but still slower than most search methods. Overall, PLMSearch searches faster than pLM-BLAST and only requires per-protein embeddings rather than larger per-residue embeddings. However, pLM-BLAST is able to generate specific sequence alignments based on residue-level information, especially local alignments."

Q 1.3 - The manuscript doesn't recognise in sufficient detail that PLMSearch is a Python-Script that wraps around existing software and language models with a few internal data-manipulation steps which would usually be done as part of a typical bioinformatics analysis workflow. The authors fail to illustrate in detail what their conceptual and technical innovation is and which aspect of their analysis wasn't available to the bioinformatics community before PLMSearch existed.

Reply: Thank you for the comments. As described above, in terms of technological innovation, PfamScan has been a well-known approach for identifying domains in sequences. Protein language models, like ESM-1b, have also been widely employed in tasks including protein structure and function prediction, protein-protein interaction prediction, and protein-phenotype association prediction. However, they are not the same as our essential components, PfamClan and SS-predictor. They are simply the foundation for PfamClan and SS-predictor (specifically, PfamScan and ESM-1b are used to preprocess protein sequences as the input to PfamClan and SS-predictor). Furthermore, PLMSearch distinguishes itself by incorporating the cutting-edge protein language model to predict similarity, thereby allowing fast searching of homologous protein sequences without structure, which is extremely valuable for protein sequence research in high-throughput circumstances. Finally, compared with other methods that also use protein language models (such as EAT and pLM-BLAST), PLMSearch also has its unique features. Compared with EAT which directly uses the Euclidean distance between embeddings, PLMSearch uses a neural

network to predict the similarity and uses the structural similarity TM-score as the ground truth, so the search results are more sensitive. Compared with pLM-BLAST, PLMSearch uses per-protein embedding instead of per-residue embedding for similarity prediction, which greatly reduces the deployment space required for the target dataset while increasing the search speed by more than four orders of magnitude. Detailed differences between PLMSearch and other methods can be examined in Table R2 (Supplementary Table 4).

Q 1.4 - *The shortcoming in explaining the innovation in this work is further magnified by the fact that benchmarking is performed and communicated highly selectively. For example, MMSeqs2 and other pairwise protein aligners were never designed and are not intended to serve for deep homology assessments, but rather focus on superior speed, because such pairwise sequence alignments can then serve as input to more sensitive approaches such as multiple sequence alignments/ Hidden Markov Models / Fold-predictors etc. However, the authors claim superior sensitivity (but vastly worse speed performance) over such pairwise approaches which is already very well known in the protein evolution and bioinformatics community and not a meaningful benchmark to begin with.*

Reply: Thank you for the comments. As described above, in our original manuscript, our baseline selection mainly referred to the latest SOTA structure search method Foldseek (2023, Nature Biotechnology, <https://www.nature.com/articles/s41587-023-01773-0>). We chose MMseqs2 as one of the baselines because MMseqs2 is a classic sequence search method and was used as the main baseline of sequence search in the Foldseek paper. We also objectively stated that MMseqs2 is efficient in the paper. Also, in general, structure search tends to have an advantage over sequence search in terms of sensitivity. Therefore, our baseline selection was mainly inclined to structure search method with higher sensitivity, which is used to be compared with the sensitivity of PLMSearch. In this revised version, for a comprehensive comparison, we have also added some cutting-edge sequence search methods as baselines, including Blastp, HHblits, EAT, and pLM-BLAST. In addition, we also added Foldseek-TM which is more sensitive than Foldseek.

Q 1.5 - *In contrast, structural search comparisons against software such as TM-Align show less sensitivity, but higher computational speed (but then no comparison is shown against HHblits or HHpred or pLM-BLAST which also aim to increase speed for deep homology detection at the cost of loss in sensitivity). Overall, the entire benchmarking setup is very selective and only a comprehensive benchmark based on all existing software (HHpred, HHblits, MMSeqs2 Profile Search, Multiple Sequence Alignment based approaches, (jack)HMMer, Profile based BLAST, ESMFold, Foldseek etc) in terms of both: speed and sensitivity would be sufficient.*

Reply: Thank you for the comments. We have added sequence search methods such as Blastp, HHblits, EAT, and pLM-BLAST as baselines for a more comprehensive comparison. In addition, we have also added Foldseek-TM, which is more sensitive than Foldseek.

Q 1.6 - *The computational speed benchmarks are very limited and rather hidden in Extended Data Table A4 and in the Supplementary Information. Computational speed should be assessed against all competitor software (see previous point), for various input dataset sizes (scaling benchmark) and for parallelisation scaling (multicore CPU and or GPU if competitor software is leveraged by GPUs rather than CPUs). In addition, it is not clear how memory consumption will limit practical use-cases and what the upper-limit boundaries of the number of sequences are that can be processes in a reasonable hardware setup (personal computer/server/HPC).*

Reply: Thank you for the comments. To ensure fairness, we used the same computing resource (a 56-core Intel(R) Xeon(R) CPU E5-2680 v4 @ 2.40 GHz and 256 GB RAM memory) when implementing

different methods in the evaluation. For many parallel methods such as pLM-BLAST and HHblits, all 56 cores are used by default. Moreover, most methods will perform data preprocessing on the target dataset in advance (such as HHblits requires pre-computing a large number of MSAs; pLM-BLAST, EAT, and PLMSearch need to pre-generate protein embeddings). However, this part of the time is not required while searching. Therefore, using GPU to generate embeddings in the preprocessing stage will not bring advantages to time statistics, because we only count the search time of different methods on the same CPU platform. In this revision, we have added the brief descriptions into the “2.2 PLMSearch searches millions of query-target pairs in seconds” as follows.

"To ensure fairness, we used the same computing resources (a 56-core Intel(R) Xeon(R) CPU E5-2680 v4 @ 2.40 GHz and 256 GB RAM server) when implementing different methods in the evaluation. For HHblits, pLM-BLAST, TM-align, and various other parallelizable methods, we utilized all 56 cores by default. Moreover, almost all methods need to preprocess the target dataset in advance. This part of the time is not required while searching, so we did not include the preprocessing time in the search time statistics."

Q 1.7 - The methods section focuses on explaining the existing tools and approaches that are incorporated into PLMSearch rather than on the actual innovation within PLMSearch that would putatively allow this software to perform better if after presenting sufficient benchmarks PLMSearch still outperforms all competitors.

Reply: Thank you for the comments. We have polished the description of our method section to highlight the innovation of PLMSearch. In addition, we have added a more comprehensive benchmark to evaluate PLMSearch, and made a rigorous major revision to our work, including comprehensively revising our experimental design, expanding the scale of the experiment, adding a variety of the latest sequence search methods as baselines, etc.

Reviewer 2

Q 2.1 *Zhu et al. presented a swift fold recognition technique, PLMsearch (and SS-predictor), employing a machine learning approach that integrates the ESM-1b protein language model, released by Meta AI (Facebook). Based on the documented results, it appears that PLMsearch outperforms the pure sequence alignment-based method, MMseqs2, in terms of fold detection. However, its performance is slightly inferior to that of the structure-based alignment tool, TM-align. Although the search speed of PLMsearch surpasses TM-align, it remains slower than the majority of state-of-the-art methods, such as Foldseek. Overall, PLMsearch seems to hold a respectable position among these methods in terms of accuracy or running speed. The concept of the SS-predictor is similar to a previous work, “A novel sequence alignment algorithm based on deep learning of the protein folding code, Bioinformatics (2021)”, but distinguishes itself by incorporating the cutting-edge protein language model into their deep learning network, which serves as a notable highlight. The deep learning-based alignment problem for sequences or sequence-structure alignment is indeed a captivating subject.*

Reply: Thanks for the encouraging comments.

Q 2.2 *Nevertheless, there exist several significant issues within this paper. In my opinion, the most crucial concern lies in the fact that the method solely reports whether two sequences (query and target) share the same fold, relying solely on a predicted TM-score, without providing any alignments. This, I believe, prevents the method from becoming a valuable tool for the scientific community. Additionally, some experimental test designs or quality assessments employed in the study are either uncommon or suffer from certain deficiencies. Please see my point-to-point comments. I hope those things will help the authors improve the quality of their work and manuscript.*

Major:

1. As previously mentioned in the general comments, it is worth noting that PLMsearch (and SS-predictor) only provides information regarding whether two sequences (query and target) are the same fold with a predicted TM-score. However, it fails to offer guidance on how these sequences should be aligned, which is provided by the Foldseek method. In protein structure prediction (such as threading) or sequence-sequence alignment (can be extended to encompass multiple sequence alignment generation problems), the primary focus is always reporting the sequence-sequence alignment or sequence-structure alignment. To illustrate this point, let’s consider an example: suppose the query corresponds to a single-domain protein that is similar to domain one of a three-domain protein target. Merely stating that the query and target share the same fold is of limited usefulness unless the method can align the query to domain one of the target and explicitly output this alignment for the user. Consequently, I highly recommend that the authors reconstruct their pipeline to include alignment as one of the output components.

Reply: Thanks for the comment. As mentioned above, guidance on how to align sequences is essential for homologous protein search. The sequence alignment answers which specific segment of the target protein the domain of the query protein is similar to.

As far as large-scale protein search is concerned, the total number of protein pairs is often as high as hundreds of millions (taking 100 query proteins locally and 500,000 target proteins in Swiss-Prot as an example, there are 50 million query-target pairs). Most of these sequence alignments are meaningless with low similarity. Therefore, the general idea is to pre-filter through some method (for example, pLM-BLAST uses COS similarity to pre-filter), and then align proteins with high similarity. To this end, we use PLMSearch for pre-filtering and perform a Smith-Waterman alignment for each pair. The reason why Smith-Waterman alignment is used by default is that PLMSearch is based on per-protein embedding. Although this ensures the high speed and high throughput of PLMSearch, it also erases the amino acid-level features. Therefore, we cannot use the average embedding of amino acids for specific sequence alignment. In this regard, we discussed “PLMAlign” in “Discussion” as follows.

"In the future, we intend to develop a deep learning method PLMAlign to improve sequence alignment. In the first step of large-scale searching, PLMSearch only considers the per-protein embedding averaged by per-residue embedding to improve efficiency and save RAM space. Subsequently, for potential homologous query-target pairs searched by PLMSearch, PLMAlign calculates the mutual attention between query-target per-residue embeddings based on the transformer network in deep learning, which is expected to explore better global and local sequence alignment results."

In addition to the Smith-Waterman alignment, we have added the parallel TM-align to the pipeline, which is available when users have the structures of the query and target proteins. We have incorporated this part into our paper, web server (<https://dmiip.sjtu.edu.cn/PLMSearch/>), and open source code (<https://github.com/maovshao/PLMSearch/blob/main/pipeline.ipynb>) (Fig. R2).

Q 2.3 2. In the "2.1 PLMsearch reaches similar..." section, during the evaluation of the PLMsearch method on the SCOPe dataset and database, the authors made comparisons with TM-align, Dali, Foldseek, and MMseqs2. However, given that this evaluation methodology resembles template detection, I would suggest that the authors also compare their method with fold recognition or threading approaches, such as HHsearch.

Since PLMsearch utilizes ESM-1b, which likely incorporates predicted 'contact/distance' information in its hidden states/layers, it would be better to also include comparisons with contact/distance-based fold recognition methods like Map-align, EigenThreader, DiscoVER, etc.

Reply: Thanks for the comments. We have added HHblits as a baseline. HHblits is presented as an HMM-HMM-based lightning-fast iterative sequence search, which extends HHsearch to enable fast and iterative sequence searches. The profile-profile alignment prefilter of HHblits reduces the number of full HMM-HMM alignments from many millions to a few thousand, making it faster than PSI-BLAST but still as sensitive as HHsearch (<https://www.nature.com/articles/nmeth.1818>). Since we have compared with methods that directly use 3D structures for alignment (Dali, CE, and TM-align), we additionally discuss alignment methods using contact/distance maps (Map-align, EigenTHREADER, and DiscoVER) in the "Introduction" as follows. This is based on the fact that structure alignment methods like TM-align are more sensitive than sequence alignment methods, including those using contact/distance maps.

"Protein structure search methods can be divided into: (1) contact/distance map-based, such as Map-align [6], EigenTHREADER [7] and DiscoVER [8];"

Q 2.4 3. In the "2.1 PLMsearch reaches similar..." section and throughout the SCOPe-based results section, the authors report the "total TPs up to the first FP" and emphasize their method could enrich more true positives in the ranking list. However, these assessments may not be particularly informative, as users without prior experience often focus solely on the first hit and tend to place trust in the result with the highest confidence score. Therefore, I suggest reporting the accuracy of the first hit, and you can refer to how this paper did such an analysis with SCOPe.

EigenThreader: <https://academic.oup.com/bioinformatics/article/33/17/2684/3611272>

Reply: Thanks for the comments. We have referred to EigenThreader [7] and added "The accuracy of the first hit" (P@1) to our evaluation as follows.

- "P@K [7]: For homologous protein search, as many queries have thousands of relevant targets, and few users will be interested in getting all of them. Precision at k (P@k) is then a useful metric (e.g., P@10 corresponds to the number of relevant results among the top 10 retrieved targets). P@K here is the mean value for all query proteins."

a

b

c

d

Fig. R2 Alignment on web server and Github. **a-b**, Sequence alignment on web server. **c**, Sequence alignment on Github. **d**, TM-align on Github.

Q 2.5 4. In the “2.1 PLMsearch reaches similar...” section, as well as in the method section “4.5.2 Evaluation based on the TM-score benchmark”, and other related sections discussing the TM-score benchmarking, the authors employed a normalized TM-score using the average length of the query and target structures (i.e., $TM\text{-align}(avg.\ length)$). However, this approach is not appropriate as the problem is query-based. During the TM-score benchmarking, the structure of the query should be known, and thus the TM-score should be normalized with the length of the query rather than the average length.

Reply: Thanks for the comments. The normalization method of TM-score has always been the key to our experimental setting. We did an exhaustive comparison of different settings in the paper and found that normalizing using the average length gave the best results (Table R4, Supplementary Table 8). The results and settings are basically consistent with the conclusions obtained from Foldseek [9] and MT-LSTM [10].

Methods	Family	Superfamily	Fold
MMseqs2			
MMseqs2(Default)	0.157	0.021	0.000
MMseqs2(Best)	0.318	0.050	0.002
Foldseek			
Foldseek(Default)	0.883	0.584	0.213
Foldseek(Best)	0.883	0.584	0.214
Foldseek-TM(Best)	0.898	0.664	0.296
TM-align			
TM-align(Default)	0.859	0.529	0.158
TM-align(Avg. score)	0.933	0.711	0.326
TM-align(Avg. length)	0.935	0.721	0.346

Table R4 Results with different settings for MMseqs2, Foldseek, and TM-align. Different settings can greatly affect sensitivity. MMseqs2(Default) and Foldseek(Default) are the default settings of the program. MMseqs2(Best), Foldseek(Best), and Foldseek-TM(Best) are the practiced parameters in the experiments of Foldseek [9]. TM-align(Default) uses the query protein length as the normalized length. TM-align(Avg. score) calculates TM-scores for both comparison directions and averages them together. TM-align(Avg. length) uses the average length of protein pairs as the normalized length. We experimented with the settings that yielded the highest sensitivity, both as a metric and as a method. The results and setting are basically consistent with the conclusions obtained from Foldseek [9] and MT-LSTM [10].

Q 2.6 5. *One issue I have identified in this paper is the excessive number of tests designed within the results section. This abundance of tests distracts the readers from focusing on the central topic. I recommend that the authors adopt a more streamlined approach by dedicating one experimental test to each section. Additionally, it would be beneficial to explain the underlying logic behind why PLMsearch is faster or superior, while providing relevant case studies. Currently, each section introduces approximately two tests (some even more), along with new concepts. This approach tends to divert the readers' attention towards understanding these concepts rather than evaluating the performance of PLMsearch.*

Reply: Thanks for the suggestions on the experimental description section. According to the suggestions, we have polished the descriptions in the “Results” section, notably reorganizing the paper’s figures and tables. To prevent distracting readers, we have reintegrated the original dispersed metrics, experimental descriptions, and experimental results and created a coherent narrative in each section. Furthermore, we have compared PLMSearch with baselines and ablation methods using the same metrics.

Q 2.7 6. *Section 2.5 could potentially be divided into separate sections to improve clarity. The first paragraph could be relocated to other relevant results sections where it directly pertains to the specific findings. Similarly, the second paragraph could be moved to the method section, as it seems to primarily explain the behavior of PLMsearch rather than presenting extensive result-related information. By reorganizing the content in this manner, the paper's structure and flow will be enhanced, allowing for a more coherent presentation of the results and methodological details.*

Reply: Thanks for the suggestions. We have reorganized Section 2.5 following the suggestions. Specifically, we moved Section 2.5 to the method section, because it was originally used to explain the similarity of SS-predictor. In this revision, we have added the corresponding descriptions into the “4.3 Similarity prediction” as follows.

"Furthermore, unlike Foldseek, which focuses on local similarity, the similarity of SS-predictor, like TM-score, focuses on global similarity (Table R5, Extended Data Table A7)."

	Query	Target	TM-score		Foldseek	SS-predictor
			Default	Avg. length	Probability	Similarity
UniProt ID	P32352	Q5HJR8	0.343	0.173	1.000	0.607
Length	222	745				
Structure						
UniProt ID	P32352	Q5U263	0.456	0.189	0.795	0.604
Length	222	1,146				
Structure						
UniProt ID	P32352	Q5RF50	0.375	0.190	0.975	0.629
Length	222	758				
Structure						
UniProt ID	P32352	Q8NYT6	0.334	0.168	0.996	0.609
Length	222	745				
Structure						

Table R5 Four protein pairs selected for the manual inspection of protein pairs. They are filtered by Foldseek but with a TM-score<0.2. TM-align(Default) uses the query protein length as the normalized length. TM-align(Avg. length) uses the average length of protein pairs as the normalized length. As reported in Foldseek's paper, Foldseek searches out these pairs because it focuses on local similarity. However, TM-align and PLMSearch focus on global similarity, so these pairs have TM-score<0.5 and similarity of SS-predictor close to 0.6.

Q 2.8 7. *Figure S3 and the histograms throughout the paper appear to lack a clear design and can be confusing for readers. I suggest either presenting different histograms in separate figures or aligning the x-axis of the histograms together for better visual clarity and comparability. This will facilitate easier interpretation and understanding of the data presented in the histograms.*

Reply: Thanks for the comments. We have removed Figure S3 because our original intention is to illustrate that we use Spark to calculate TM-align in parallel with multiple threads. Now the parallelized

TM-align in our pipeline has been changed to use a more convenient implementation based on multiprocessing and can output specific alignments at the same time. In addition, other histograms such as Fig. 4 and Extended Data Fig. A2 have been adjusted to make the presentation of the paper clearer.

Q 2.9 8. In the “4.4.2 Swiss-Prot” section, line 498, 5 proteins are too few to test!

Reply: Thanks for the comments. We have set the number of queries for the Swiss-Prot to Swiss-Prot search test to 50. In this revision, we have added the corresponding descriptions into the “4.4.2 Swiss-Prot” as follows.

"Subsequently, we randomly selected 50 queries from Swiss-Prot and 50 queries from SCOPe40-test as query proteins (a total of 100 query proteins) and searched for 430,140 target proteins in Swiss-Prot. Therefore, a total of 43,014,000 query-target pairs were tested. The search test for 50 query proteins from Swiss-Prot and SCOPe40-test are called "Swiss-Prot to Swiss-Prot" and "SCOPe40-test to Swiss-Prot" respectively."

Q 2.10 9. Still, in the “4.4.2 Swiss-Prot” section, it is crucial to report the redundancy of the five Swiss-Prot proteins with the SCOPe training dataset. The authors should employ non-redundant Swiss-Prot sequences when testing their method against the SCOPe40-train dataset.

Reply: Thanks for the comments. We have performed a redundancy filter for Swiss-Prot. In this revision, we have added the corresponding descriptions into the “4.4.2 Swiss-Prot” as follows.

"In order to avoid possible data leakage issues, like SCOPe40, we used 0.4 sequence identity as the threshold to filter homologs in Swiss-Prot from the training set. Specifically, we used the previously screened 498,659 proteins as query proteins and SCOPe40-train as the target dataset. We first pre-filtered potential homologous protein pairs with MMseqs2 and calculated the sequence identity between all these pairs. The query protein from Swiss-Prot will be discarded if the sequence identity between the query protein and any target protein is greater than or equal to 0.4. Finally, 68,519 proteins were deleted via homology filtering, leaving 430,140 proteins in Swiss-Prot, which we employed in our experiments. We also studied the max sequence identity of each protein in Swiss-Prot relative to SCOPe40-train (Fig. R3, Supplementary Fig. 4) and found that the vast majority of them were below 0.3."

Q 2.11 10. What is “- >” in Figure 5? Please explain more about this figure, as its current representation is causing some confusion.

Reply: Thanks for the comments. We have redesigned the ablation experiment and removed this figure.

Q 2.12 11. All pipeline and explanation figures in this paper utilize a DNA logo instead of a protein sequence logo. It is recommended to replace the DNA logo with a protein sequence logo to align with the focus of the paper.

Reply: Thank you for your careful observation. We have corrected Main text Fig. 1, Fig. 3, and Fig 5 accordingly. See Fig. R4 (Main text Fig. 1) as an example.

Fig. R3 The data distribution of max sequence identity of each protein in SCOPe40-test and Swiss-Prot against SCOPe-train. The majority of the maximum sequence identity is between 0.2 and 0.3. The sequence identity difference between their data is significantly bigger than that of pure random division, especially for the SCOPe40-test, which is the major test data, since the domains in SCOPe40-test belong to different folds with all domains in SCOPe40-train.

Q 2.13 *Minor:*

1. Line 17 on page 2, it should be “HH-suite” instead of “Hh-suite”
2. Line 18-21 on page 2, the sentence “These methods are based on the idea that homologous protein pairs with high sequence similarity often share similar molecular and cellular functions and structures.” does not make sense here. It seems that the authors intended to summarize the methodology of sequence alignment methods such as BLAST and MMseqs2. However, the sentence does not effectively convey that meaning. Please revise it.

Reply: Thanks for the comments. We have modified the presentation in “Introduction” as follows.

"Due to the low cost and large scale of sequence data, the most widely used homologous protein search methods are based on sequence similarity, such as MMseqs2 [12], BLASTp [13], and Diamond [14]. Despite the success of homology inference based on sequence similarity, it remains challenging to detect distant evolutionary relationships from sequences only [15]. Sequence profiles and profile hidden Markov models (HMMs) are condensed representations of multiple sequence alignment (MSAs) that specify for each position the probability of observing each of the 20 amino acids in evolutionarily related proteins. When the sequence identity is lower than 0.3, methods based on profile HMMs such as HMMER [16], HHsearch [17] and HHblits [18, 19], are currently the best tools for homologous protein search."

Fig. R4 Overview of the PLMSearch pipeline. **a**, PfamClan. Initially, PfamScan [11] identifies the Pfam clan domains of the query protein sequences, which are depicted in different color blocks. Subsequently, PfamClan searches the target dataset for proteins sharing the same Pfam clan domain with the query proteins. Notably, the last query protein lacks any Pfam clan domain, and therefore, its all pairs with target proteins are retained. **b**, Similarity prediction. The protein language model generates deep sequence embeddings for query and target proteins. Subsequently, SS-predictor predicts the similarity of all query-target pairs. **c**, Search result. Finally, PLMSearch selects the similarity of the protein pairs pre-filtered by PfamClan, sorts these protein pairs based on their predicted similarity, and outputs the search results for each query protein separately.

Reviewer 3

Q 3.1 *Liu et al. present PLMSearch, which allows fast searching of homologous protein sequences without structure. In the data rich era of biology this is an important problem to solve. This paper could be of general interest if issues surrounding data leakage, a common problem in this field, can be addressed.*

Reply: Thanks for the encouraging comments.

Q 3.2 # Major points

1. *The strategy of randomly splitting SCOPe40 8:2 like in Foldseek is likely not sufficient to avoid data leakage for models that are directly training on the structural similarity. It means that sequences with 39% sequence identity, indicating near-certain homology, could be in both the training and the test set, which will give inflated results compared to unseen sequences. This can be studied by plotting the distribution of “closest sequence ID to the training set” for the test set sequences. If this indicates significant overlap of the sets, it may be necessary to exclude some sequences from testing or retrain with a more stringent training set.*

Reply: Thanks for the comments. We have provided additional details on how our data is split. Furthermore, according to the suggestions, we also counted the max sequence identity of each protein in SCOPe40-test relative to SCOPe40-train. In fact, as we stated in “4.4.1 SCOPe40”, the difference between training and testing data is much larger than that of pure random division, and most of the max sequence identity is between 0.2 and 0.3. The details are as follows.

"As done in Foldseek, domains from SCOPe40 were split 8:2 by fold into SCOPe40-train and SCOPe40-test, and then domains with a single chain were reserved. We trained SS-predictor on SCOPe40-train and performed tests on SCOPe40-test as a benchmark. It is worth mentioning that each domain in SCOPe40-test belongs to a different fold from all domains in SCOPe40-train, so the difference between training and testing data is much larger than that of pure random division. We also studied the max sequence identity of each protein in SCOPe40-test relative to SCOPe40-train (Fig. R3, Supplementary Fig. 4). From the figure, we can draw a similar conclusion that the sequences in SCOPe40-test and SCOPe40-train are quite different, and most of the max sequence identity is between 0.2 and 0.3."

Q 3.3 2. *For the Swiss-Prot dataset, it appears that no filtering of homologs from the training set was done at all. This explains the high performance compared to Foldseek - there is likely significant data leakage. This can be studied using the same approach as above. Limiting the Swiss-Prot set to those with less than 30% sequence ID to sequences used for training may be sufficient here.*

Reply: Thanks for the comments. We have performed a redundancy filter for Swiss-Prot. In this revision, we have added the corresponding descriptions into the “4.4.2 Swiss-Prot” as follows.

"In order to avoid possible data leakage issues, like SCOPe40, we used 0.4 sequence identity as the threshold to filter homologs in Swiss-Prot from the training set. Specifically, we used the previously screened 498,659 proteins as query proteins and SCOPe40-train as the target dataset. We first pre-filtered potential homologous protein pairs with MMseqs2 and calculated the sequence identity between all these pairs. The query protein from Swiss-Prot will be discarded if the sequence identity between the query protein and any target protein is greater than or equal to 0.4.

Finally, 68,519 proteins were deleted via homology filtering, leaving 430,140 proteins in Swiss-Prot, which we employed in our experiments. We also studied the max sequence identity of each protein in Swiss-Prot relative to SCOPe40-train (Fig. R3, Supplementary Fig. 4) and found that the vast majority of them were below 0.3."

Q 3.4 3. *PLMSearch has considerably better performance than SS-predictor. This is a problem when extrapolating the performance of the model to new, unclassified sequences that are not in Pfam; they will presumably not find any Pfam matches and the performance will be equivalent to SS-predictor. These new cases are the ones of most interest when looking for remote homology, since anything with Pfam matches already has its homologs identified. This effect is hard to benchmark since the SCOP test set tends to be well-annotated in Pfam. However it should be discussed thoroughly.*

Reply: Thanks for the comments. These new, unclassified sequences that cannot scan out any Pfam domain are often very important for innovative protein research. We have discussed search results for new proteins without any Pfam domain in "2.4 Ablation experiments: PfamClan and SS-predictor make PLMSearch more robust" as follows.

"To evaluate PLMSearch without the PfamClan component, we added an additional search test that only included query proteins that failed to scan any Pfam domain. This corresponds to the new and unclassified proteins in the real scene, and these sequences are often very important for some innovative research. Specifically, we screened a total of 123 proteins that failed to scan any Pfam domain from 2,207 queries in the SCOPe40-test. We used MAP and P@K to evaluate the search results of these 123 proteins (Fig. R5 (Supplementary Fig. 2), Table R6 (Supplementary Table 2)). As expected, the performance of PLMSearch is exactly the same as that of SS-predictor, because PfamClan does not filter out any protein pairs, whereas PLMSearch still produces relatively sensitive search results (MAP = 0.577, P@1 = 0.813, P@10 = 0.726)."

Q 3.5 4. *It would be good to mention and compare to EAT (<https://github.com/Rostlab/EAT>) which is also a sequence-based search tool that uses contrastive learning based on structural classification. It is difficult to benchmark other trained approaches without data leakage though.*

Reply: Thanks for the comments. We have added EAT as a baseline for a more comprehensive comparison. In this revision, we have also added the corresponding descriptions into the "Introduction" as follows.

"Embedding-based annotation transfer (EAT) [20] uses Euclidean distance between vector representations (ProtT5 embeddings) of proteins to transfer annotations from a set of labeled lookup protein embeddings to query protein embeddings;"

Q 3.6 5. *Is it possible to run an ablation using only the PfamClan step, i.e. ranking by the PfamClan score and not re-ranking by SS-predictor? This would give information on how much SS-predictor contributes to the performance of PLMSearch over a simple HMM search.*

Reply: Thanks for the comments. As described in the experimental analysis, PfamClan can only classify protein pairs as two types, in which 0 and 1 represent not sharing the same Pfam domain and sharing the same Pfam domain, respectively. PfamClan itself cannot predict similarity and rank. In the ablation experiment and the new protein experiment, we have made a detailed analysis of the contribution of PfamClan and the performance of using only SS-predictor, and found that PfamClan can eliminate the

Fig. R5 Evaluation on new proteins. New proteins refer to new, unclassified sequences without any Pfam annotation. They are often necessary for innovative protein research. In the all-versus-all search test on the SCOPe40-test dataset, we counted the MAP, P@1, and P@10 metrics with only new proteins as queries (123 in total). The full results are as follows: **a**, Baselines. **b**, Ablation methods. **c**, Other baselines. Table R6 records the specific values of each metric. As expected, on the queries without any Pfam annotation, the performance of PLMSearch is equivalent to SS-predictor, but still maintains a high sensitivity.

Methods	MAP	P@K	
	MAP	P@1	P@10
Sequence search			
MMseqs2	0.107	0.495	0.137
Blastp	0.133	0.585	0.260
HHblits	0.297	0.886	0.575
EAT	0.318	0.788	0.552
pLM-BLAST	0.682	0.943	0.783
Structure search — structural alphabet			
3D-BLAST-SW	0.379	0.764	0.560
CLE-SW	0.392	0.788	0.557
Foldseek	0.521	0.878	0.712
Foldseek-TM	0.562	0.886	0.723
Structure search — structural alignment			
CE	0.576	0.869	0.699
Dali	0.636	0.910	0.772
TM-align	0.770	0.951	0.799
Our methods			
Euclidean	0.371	0.796	0.609
COS	0.376	0.796	0.616
SS-predictor (w/o COS)	0.568	0.813	0.717
SS-predictor	0.577	0.813	0.726
PLMSearch	0.577	0.813	0.726

Table R6 Evaluation on new proteins. New proteins refer to new, unclassified sequences without any Pfam annotation. The definition of MAP, P@K is detailed in “Metrics” Section. The highest value achieved for each metric is highlighted in bold.

top-ranked FPs to help improve sensitivity. Only using SS-predictor also obtains relatively sensitive search results as shown in Table R6 (Supplementary Table 2).

Q 3.7 # Minor points

1. Mention in the intro the reason why searching for homology with structures is effective, i.e. that structure is more conserved than sequence. Also describe briefly what protein language models are, their output ($n \times d$) and how they are trained.

Reply: Thanks for the comments. We have revised relevant sections in the “Introduction” and “Methods” as follows.

"In cases of highly distant evolutionary relationships, sequences may have diverged to such an extent that we can no longer detect their relatedness. Since structures diverge much more slowly than sequences, detecting similarity between protein structures by 3D superposition provides higher sensitivity [21]."

"At the same time, protein language models (PLMs) such as ESMs [22--24] and ProtTrans [25] only take protein sequences as input, trained on hundreds of millions of unlabeled protein sequences using self-supervised tasks such as masked amino acid prediction. PLMs perform well in various downstream tasks [26], especially in structure-related tasks like secondary structure prediction and contact prediction [27]."

"The input protein sequences are first sent to the protein language model (ESM-1b here) to generate per-residue embeddings ($m \times d$, where m is the sequence length and d is the dimension of the vector), and the per-protein embedding ($1 \times d$) is obtained through the average pooling layer."

Q 3.8 2. *Unless I missed it, the origin of the name SS-predictor is not given. It could be given when the name is first introduced to avoid confusion with Secondary Structure.*

Reply: Thanks for the comments. We have explained SS-predictor when we first introduced it as follows.

"(2) SS-predictor (Structural Similarity predictor) predicts the similarity between all query-target pairs with embeddings generated by the protein language model."

Q 3.9 3. *Figure 3 might be clearer with different colours for recalled and missed pairs.*

Reply: Thanks for the comments. We have emphasized "For recalled pairs (left) and missed pairs (right) in each subplot" in the caption of the figure to explain that all the points in the left picture are missed pairs, while all the points in the right picture are missed pairs.

Q 3.10 4. *In Figure 4 the E-value for MMseqs2 is given as "None" but this is ambiguous with a low E-value. Better to replace it with "No hit" or similar.*

Reply: Thanks for the comments. We have changed "None" to "No hit" accordingly.

Q 3.11 5. *It would be useful to scale the run time up to larger databases, for example estimating the run time for searching UniRef90 and the AlphaFold database.*

Reply: Thanks for the comments. In fact, in order to further expand the influence of PLMSearch, we have added PDB (680K proteins) and UniRef50 (53.6M proteins, which is nearly 100 times that of Swiss-Prot) as one of the target datasets on the web server. At the same time, we have evaluated the average time required for a single query in Table R1 (Extended Data Table A4). We have revised relevant sections in the "Target datasets on web server" as follows.

"We currently have the following four target datasets for users to search: (1) Swiss-Port (568K proteins) [1], the original dataset without de-redundancy; (2) PDB (680K proteins) [2--4]; (3) UniRef50 (53.6M proteins) [1]. UniRef50 is built by clustering UniRef90 seed sequences that have at least 50% sequence identity to and 80% overlap with the longest sequence in the cluster; (4) Self (the query dataset itself). Searching a query on Swiss-Prot and UniRef50 takes approximately 0.03 min and 0.5 min respectively (Table R1)."

References

- [1] Consortium, T.U.: UniProt: the Universal Protein Knowledgebase in 2023. *Nucleic Acids Research* **51**(D1), 523–531 (2022)
- [2] Berman, H.M., Westbrook, J., Feng, Z., Gilliland, G., Bhat, T.N., Weissig, H., Shindyalov, I.N., Bourne, P.E.: The Protein Data Bank. *Nucleic Acids Research* **28**(1), 235–242 (2000)
- [3] Burley, S.K., Bhikadiya, C., Bi, C., Bittrich, S., Chen, L., Crichlow, G.V., Christie, C.H., Dalenberg, K., Di Costanzo, L., Duarte, J.M., Dutta, S., Feng, Z., Ganesan, S., Goodsell, D.S., Ghosh, S., Green, R.K., Guranović, V., Guzenko, D., Hudson, B.P., Lawson, C., Liang, Y., Lowe, R., Namkoong, H., Peisach, E., Persikova, I., Randle, C., Rose, A., Rose, Y., Sali, A., Segura, J., Sekharan, M., Shao, C., Tao, Y.-P., Voigt, M., Westbrook, J., Young, J.Y., Zardecki, C., Zhuravleva, M.: RCSB Protein Data Bank: powerful new tools for exploring 3D structures of biological macromolecules for basic and

applied research and education in fundamental biology, biomedicine, biotechnology, bioengineering and energy sciences. *Nucleic Acids Research* **49**(D1), 437–451 (2020)

- [4] Sehnal, D., Bittrich, S., Deshpande, M., Svobodová, R., Berka, K., Bazgier, V., Velankar, S., Burley, S.K., Koča, J., Rose, A.S.: Mol* Viewer: modern web app for 3D visualization and analysis of large biomolecular structures. *Nucleic Acids Research* **49**(W1), 431–437 (2021)
- [5] Kaminski, K., Ludwiczak, J., Pawlicki, K., Alva, V., Dunin-Horkawicz, S.: plm-blast – distant homology detection based on direct comparison of sequence representations from protein language models. *bioRxiv* (2023)
- [6] Ovchinnikov, S., Park, H., Varghese, N., Huang, P.-S., Pavlopoulos, G.A., Kim, D.E., Kamisetty, H., Kyrpides, N.C., Baker, D.: Protein structure determination using metagenome sequence data. *Science* **355**(6322), 294–298 (2017)
- [7] Buchan, D.W.A., Jones, D.T.: EigenTHREADER: analogous protein fold recognition by efficient contact map threading. *Bioinformatics* **33**(17), 2684–2690 (2017)
- [8] Bhattacharya, S., Roche, R., Moussad, B., Bhattacharya, D.: Discover: distance- and orientation-based covariational threading for weakly homologous proteins. *Proteins: Structure, Function, and Bioinformatics* **90**(2), 579–588 (2022)
- [9] van Kempen, M., Kim, S.S., Tumescheit, C., Mirdita, M., Lee, J., Gilchrist, C.L., Söding, J., Steinegger, M.: Fast and accurate protein structure search with foldseek. *Nature Biotechnology*, 1–4 (2023)
- [10] Bepler, T., Berger, B.: Learning the protein language: Evolution, structure, and function. *Cell systems* **12**(6), 654–669 (2021)
- [11] Mistry, J., Bateman, A., Finn, R.D.: Predicting active site residue annotations in the pfam database. *BMC bioinformatics* **8**, 298 (2007)
- [12] Steinegger, M., Söding, J.: Mmseqs2 enables sensitive protein sequence searching for the analysis of massive data sets. *Nature biotechnology* **35**(11), 1026–1028 (2017)
- [13] Altschul, S.F., Gish, W., Miller, W., Myers, E.W., Lipman, D.J.: Basic local alignment search tool. *Journal of Molecular Biology* **215**(3), 403–410 (1990)
- [14] Buchfink, B., Reuter, K., Drost, H.-G.: Sensitive protein alignments at tree-of-life scale using diamond. *Nature methods* **18**(4), 366–368 (2021)
- [15] Mahlich, Y., Steinegger, M., Rost, B., Bromberg, Y.: HFSP: high speed homology-driven function annotation of proteins. *Bioinformatics* **34**(13), 304–312 (2018)
- [16] Eddy, S.R.: Accelerated profile hmm searches. *PLoS computational biology* **7**(10), 1002195 (2011)
- [17] Söding, J.: Protein homology detection by HMM–HMM comparison. *Bioinformatics* **21**(7), 951–960 (2004)
- [18] Remmert, M., Biegert, A., Hauser, A., Söding, J.: Hhblits: lightning-fast iterative protein sequence searching by hmm-hmm alignment. *Nature methods* **9**(2), 173–175 (2012)
- [19] Steinegger, M., Meier, M., Mirdita, M., Vöhringer, H., Haunsberger, S.J., Söding, J.: Hh-suite3 for

- fast remote homology detection and deep protein annotation. *BMC bioinformatics* **20**(1), 1–15 (2019)
- [20] Heinzinger, M., Littmann, M., Sillitoe, I., Bordin, N., Orengo, C., Rost, B.: Contrastive learning on protein embeddings enlightens midnight zone. *NAR Genomics and Bioinformatics* **4**(2) (2022). lqac043
- [21] Illergård, K., Ardell, D.H., Elofsson, A.: Structure is three to ten times more conserved than sequence—a study of structural response in protein cores. *Proteins: Structure, Function, and Bioinformatics* **77**(3), 499–508 (2009)
- [22] Rives, A., Meier, J., Sercu, T., Goyal, S., Lin, Z., Liu, J., Guo, D., Ott, M., Zitnick, C.L., Ma, J., *et al.*: Biological structure and function emerge from scaling unsupervised learning to 250 million protein sequences. *Proceedings of the National Academy of Sciences* **118**(15), 2016239118 (2021)
- [23] Meier, J., Rao, R., Verkuil, R., Liu, J., Sercu, T., Rives, A.: Language models enable zero-shot prediction of the effects of mutations on protein function. *Advances in Neural Information Processing Systems* **34**, 29287–29303 (2021)
- [24] Lin, Z., Akin, H., Rao, R., Hie, B., Zhu, Z., Lu, W., dos Santos Costa, A., Fazel-Zarandi, M., Sercu, T., Candido, S., *et al.*: Language models of protein sequences at the scale of evolution enable accurate structure prediction. *bioRxiv* (2022)
- [25] Elnaggar, A., Heinzinger, M., Dallago, C., Rehawi, G., Wang, Y., Jones, L., Gibbs, T., Feher, T., Angerer, C., Steinegger, M., *et al.*: Prottrans: Toward understanding the language of life through self-supervised learning. *IEEE transactions on pattern analysis and machine intelligence* **44**(10), 7112–7127 (2021)
- [26] Hu, M., Yuan, F., Yang, K.K., Ju, F., Su, J., Wang, H., Yang, F., Ding, Q.: Exploring evolution-based &-free protein language models as protein function predictors. *arXiv preprint arXiv:2206.06583* (2022)
- [27] Rao, R., Bhattacharya, N., Thomas, N., Duan, Y., Chen, P., Canny, J., Abbeel, P., Song, Y.: Evaluating protein transfer learning with tape. *Advances in neural information processing systems* **32** (2019)

Reviewer #1 (Remarks to the Author):

I very much appreciate that the authors have taken the time to carefully address most of my concerns and comments.

While many issues have been resolved, there remains an important clarification for the claim "1) pLM-BLAST needs to use per-residue embeddings for global and local alignments, which means that it needs to retain the embeddings of all amino acids for each protein in the target dataset ($N \times L \times D$, N represents the number of proteins, L represents the length of the protein, and D represents the embedding dimension). In contrast, PLMsearch only needs to retain per-protein embeddings as the target dataset ($N \times 1 \times D$).".

How does PLMsearch detect local alignments/conservation in the target dataset if entire per-protein embeddings are stored? The reason why approaches such as pLM-BLAST store per-residue embeddings is to be able to find both: local and global alignments/preservation. All provided benchmarks should consider this important aspect.

Since this is critical to the presented method itself, it would be crucial to address this remaining concern in detail.

After addressing this aspect I can support a editorial decision to accept this manuscript for publication.

Reviewer #2 (Remarks to the Author):

In the revised manuscript, Zhu et al. addressed most of my former concerns in a relatively satisfying manner. However, I still believe that several major issues have not been adequately answered. Additionally, newly added benchmark results demonstrate that PLMsearch is not as effective as some sequence-based control methods (PLM-BLAST and HHblits) in the context of new protein fold detection. This suggests the possibility of overfitting with known data. Furthermore, the results presented in the main text appear selective, with data indicating the disadvantages of PLMsearch being relegated to supplementary information and not discussed in the main text. This raises concerns about the current state of the revised text and its suitability for publication. I would recommend that the authors carefully address the following critical issues. Please refer to my point-to-point comments.

Major:

1. In the previous review round, I raised a critical concern regarding PLMsearch's inability to generate sequence-sequence or sequence-structure alignments. This capability is often expected by users, and simply stating that sequences are from the same fold without providing alignment information is not meaningful. I agree with the authors that generating alignments for all query-target pairs in the entire database is expensive and not very useful. Therefore, generating alignments only for the top-ranked candidates is sufficient. However, the method used by the authors to generate alignments, specifically the use of the Smith-Waterman alignment, is not ideal. Smith-Waterman alignment is typically effective for homologous pairs with a sequence identity (SID) greater than 70%, but it is not suitable for remote homologous pairs with SID less than 30%. Taking this approach might risk losing the advantages of PLMsearch. Even traditional profile-based methods such as PSI-BLAST or profile-HMM-based approaches like HHblits and HHsearch are more sensitive in both homologous and remote homologous alignments.

To address this issue, the authors could consider a more suitable approach, i.e., what PLM-BLAST did in their work, which integrates residue-level embeddings from ESM with residue-level profiles from either HHblits or PSI-BLAST profiles and applies these to dynamic programming-based alignment methods, resulting in a more robust alignment. As indicated by your benchmark results, PLM-BLAST demonstrates strong and robust performance. Note that simply relying on PLM-BLAST without any additional implementations may not be appreciated in the next round of revisions, as

it may give the impression that your pipeline only consists of a combination of third-party tools.

2. The new benchmark results comparing PLM-BLAST, HHblits, and PLMsearch (Extended Data Table A1 and Supplementary Table 2) indicate that PLMsearch does not outperform PLM-BLAST. Specifically, for "new proteins," PLMsearch performs significantly worse than PLM-BLAST (P@1 values of 0.813 and 0.943, respectively). This suggests that PLMsearch may have a disadvantage in detecting homologous proteins with new or challenging folds and may be overfitting to known or common folds. The authors appear to selectively place these disadvantageous data in supplementary materials and avoid discussing performance differences between PLMsearch and PLM-BLAST in the main text, which may not effectively address the issues with PLMsearch.

The authors attribute the poor performance of PLMsearch on "new proteins" to the PfamClan method, which relies on PfamScan, a third-party method. However, since your pipeline integrates PfamClan as one component, users are likely to be concerned about the overall performance of your pipeline and may not differentiate between PfamClan and the SS-predictor. If the authors believe that PfamClan is not suitable for detecting remote homologous "new proteins," they could consider either replacing it with their own method or re-tuning PfamClan. From a user's perspective, they may prefer to use PLM-BLAST over PLMsearch since PLM-BLAST is comparable or slightly better than PLMsearch for general fold proteins and significantly better for "new proteins." Additionally, even for general folds (Extended Data Table A1), the SS-predictor (developed by the authors) performs significantly worse than PLM-BLAST. After integrating the third-party method, PfamScan, into the meta-method PLMsearch, it only achieves comparable results with PLM-BLAST.

Focusing on limitations in the design of the PLM-BLAST method and ignoring its robust and strong performance is not an effective way to make a comparison and highlight the authors' method. Moreover, some of the limitations mentioned by the authors regarding PLM-BLAST may actually be considered advantages by some users. For example, PLM-BLAST can generate alignments using residue-level embeddings from a protein language model.

3. The current PLMsearch pipeline comprises three main components:

(i) PfamClan, which is primarily based on PfamScan, a third-party method for quickly scanning coordinates for remote homologous proteins. This component plays a major role in saving time during the search.

(ii) The SS-predictor, which is the core component developed by the authors. It predicts similarity scores between the query and coordinates from (i) and subsequently re-ranks them.

(iii) Generation of alignments using the Smith-Waterman algorithm based on BLOSUM62 for sequence-sequence pairs, which is also a third-party method (a module from Biopython).

As mentioned, (i) presents a critical issue due to PfamClan's weakness in detecting remote homologous "new proteins" with new and challenging folds, causing PLMsearch to perform much worse than sequence-based methods like PLM-BLAST and HHblits. Additionally, (iii) employing the classical Smith-Waterman algorithm is not well-suited for remote homologous alignment, which contradicts the main objective of developing a pipeline for remote homologous protein detection.

Regarding the core component (ii) developed by the authors, it does not exhibit significant differences from some older machine learning-based fold recognition methods, except for the integration of the protein language model. Examples from the following papers have quite similar ideas to SS-predictor:

- Pietro Di Lena, Pierre Baldi, "Fold recognition by scoring protein maps using the congruence coefficient," *Bioinformatics*, Volume 37, Issue 4, February 2021, Pages 506–513.

- Bin Liu, Chen-Chen Li, Ke Yan, "DeepSVM-fold: protein fold recognition by combining support vector machines and pairwise sequence similarity scores generated by deep learning networks,"

Briefings in Bioinformatics, Volume 21, Issue 5, September 2020, Pages 1733–1741.

I would highly recommend that the authors focus on making novel contributions to components (i) and (iii), aiming to strengthen the entire pipeline's performance for both general and challenging fold (new protein) recognition.

Reviewer #3 (Remarks to the Author):

I believe that the authors have mostly addressed my comments and the manuscript is ready for publication.

Response Letter

Dear Reviewers,

We greatly appreciate your time and efforts in reviewing the manuscript. We thank the reviewers for the highly pertinent and helpful comments. In the revision, we have conducted additional experiments and modified our manuscript accordingly by carefully considering the reviewers' suggestions and comments.

Here we summarize the major changes made in the revised version of our manuscript.

1 Develop PLMAlign for alignments and alignment scores

Unlike "Search method" such as PLMSearch, PLMAlign is an "Alignment method" (Supplementary Table 5). In the revision, we have added the brief descriptions in "PLMAlign" Section. We have also added a detailed introduction in "PLMAlign pipeline" Supplementary Section.

Besides, we have evaluated the search results with alignment scores generated by PLMAlign and the alignment accuracy of PLMAlign on the independent datasets Malisam and Malidup. A detailed discussion of these evaluations is provided in the subsequent one-by-one response.

2 Pfam dataset update

We have recently updated the Pfam dataset to its latest version (Pfam36.0, 2023-09-12) and reused PfamScan to scan the protein domains. With the inclusion of the latest Pfam dataset, we observed a reduction in "New proteins" (from 123 to 110) and a slight enhancement in the performance of PLMSearch (about 1% overall). Moving forward, we are committed to regularly updating PLMSearch's Pfam dataset to ensure its ongoing accuracy and relevance.

3 Add CATHS40 as training data and summarize all datasets

The SCOPe40-train dataset includes 8,953 proteins and TM-scores for all protein pairs were calculated for training. As the majority of these pairs had TM-scores below 0.5, only 504,553 pairs (among 80,156,209 in total) had a TM-score above 0.5 for model training. To enhance the model's generality, we supplemented it with high-quality protein pairs extracted from the curated CATH domain dataset [1, 2]. We also removed proteins with a sequence identity higher than 0.4 with the test dataset to avoid information leakage (Supplementary Fig. 4). By adding training data from CATHS40, we improved the performance of SS-predictor, leading to an improvement in the performance of PLMSearch as well (about 3% overall, in both "General" and "New" proteins). In this revision, we have added the detailed descriptions in "CATHS40" Section.

In addition, we have summarized the sizes and uses of the datasets mentioned in our paper in Extended Data Table A8, including training, test, target datasets on web server, and evaluation on remote homology alignment.

4 Update the PLMSearch web server and code, alongside publish the PLMAlign web server and code

- PLMSearch web server : <https://dmiip.sjtu.edu.cn/PLMSearch>
- PLMSearch source code : <https://github.com/maovshao/PLMSearch>
- PLMAlign web server : <https://dmiip.sjtu.edu.cn/PLMAlign>
- PLMAlign source code : <https://github.com/maovshao/PLMAlign>

Without compromising search efficiency, the PLMSearch server will now furnish PLMAlign results for the Top-10 targets (Fig. R1a-b). Additionally, the PLMAlign server and open-source code have been released for more alignment needs (Fig. R1c-d). We believe PLMSearch and PLMAlign are ready to serve researchers now.

In the following, we present one-by-one responses to the reviewers' comments.

a **PLMSearch**

b **Result** Job ID: 1702307338, Method: PLMSearch, Target dataset: Swiss-Prot

Target	Similarity	UniProt Link	AlphaFold Link	Sequence & Structure	Protein Name	Organism	Taxonomic Identifier
P32352	1.0000	P32352	P32352	PA32A_PUB (pLDDT=98)	C-8 sterol isomerase ERG2	Saccharomyces cerevisiae (strain ATCC 204508 / S288c)	559262 NCBI

c **PLMAlign**

d **Result** Job ID: 1702283909

Target	Score
P32352	33.6237

Fig. R1 Web server of PLMSearch and PLMAlign **a**, Homepage of the PLMSearch web server. **b**, Preview of the result of the PLMSearch web server. **a**, Homepage of the PLMAlign web server. **b**, Preview of the result of the PLMAlign web server.

Reviewer 1

Q 1.1 I very much appreciate that the authors have taken the time to carefully address most of my concerns and comments.

While many issues have been resolved, there remains an important clarification for the claim "1) pLM-BLAST needs to use per-residue embeddings for global and local alignments, which means that it needs to retain the embeddings of all amino acids for each protein in the target dataset ($N \times L_i \times D$, N represents the number of proteins, L_i represents the length of the protein, and D represents the embedding dimension). In contrast, PLMSearch only needs to retain per-protein embeddings as the target dataset ($N \times 1 \times D$)."

How does PLMSearch detect local alignments/conservation in the target dataset if entire per-protein embeddings are stored? The reason why approaches such as pLM-BLAST store per-residue embeddings is to be able to find both: local and global alignments/preservation. All provided benchmarks should consider this important aspect.

Since this is critical to the presented method itself, it would be crucial to address this remaining concern in detail.

After addressing this aspect I can support a editorial decision to accept this manuscript for publication.

Reply: Thanks for the encouraging comments. As expected, search methods like PLMSearch, relying solely on per-protein embeddings, lack residue-level features in their input data, making it unable to provide any suggestions for alignment. In Supplementary Table 5, we provide a detailed explanation about the differences between "Search methods" and "Alignment methods". Accordingly, we have modified the description in "Discussion" Section.

To address this limitation, as outlined earlier, we developed PLMAlign. pLM-BLAST [3] extends the concept of Smith-Waterman/Needleman-Wunsch algorithm (SW/NW) by replacing invariant substitution matrices with per-residue similarities between protein embeddings. Inspired by pLM-BLAST, PLMAlign calculates the substitution matrix by dot producting the per-residue embeddings of the query-target protein pair. The substitution matrix is then used in the SW/NW algorithm to perform local/global alignment, and the algorithm is accelerated through the linear gap penalty. In the revision, we have added the brief descriptions in "PLMAlign" Section. We have also added a detailed introduction in "PLMAlign pipeline" Supplementary Section. The differences between the SW/NW algorithm, pLM-BLAST, and PLMAlign are discussed in further detail in Supplementary Table 7.

Reviewer 2

Q 2.1 In the revised manuscript, Zhu et al. addressed most of my former concerns in a relatively satisfying manner. However, I still believe that several major issues have not been adequately answered. Additionally, newly added benchmark results demonstrate that PLMsearch is not as effective as some sequence-based control methods (PLM-BLAST and HHblits) in the context of new protein fold detection. This suggests the possibility of overfitting with known data. Furthermore, the results presented in the main text appear selective, with data indicating the disadvantages of PLMsearch being relegated to supplementary information and not discussed in the main text. This raises concerns about the current state of the revised text and its suitability for publication. I would recommend that the authors carefully address the following critical issues. Please refer to my point-to-point comments.

Reply: Thanks for the comments. We are pleased that most of your former concerns have been resolved. In the following, I will answer one by one the several major issues that have not been adequately answered before.

(1) The performance gap between PLMSearch/SS-predictor ("Search method") and pLM-BLAST ("Alignment method") is common because of the difference in the amount of information contained in

their inputs (Supplementary Table 5). Specifically, per-protein embeddings are three orders of magnitude smaller than per-residue embeddings, but do not contain residue-level information. Compared with SS-predictor, PLMSearch greatly improves performance through PfamClan pre-filtering. But this is not enough to make up for the information gap between per-protein embedding and per-residue embedding, especially on “New proteins” where Pfam information is completely empty.

Considering the scenario of searching extremely large datasets like UniRef50, SS-predictor and PLM-Search will still only use per-protein embedding for search to provide the best tradeoff between accuracy and speed. If more refined sorting result is needed, PLMAlign is provided to use alignment scores to rerank the search results of SS-predictor and PLMSearch (as what SS-predictor + PLMAlign and PLM-Search + PLMAlign do). The performance gap has nothing to do with overfitting with known data, which we will explain in detail later.

(2) The experiments on the “New proteins” dataset were added for the first time after the last round of review. Due to the limitation on the number of figures and tables in the main text, we unified the figures and tables of the “New proteins” experiment results in the supplementary material. However, we discussed the results on the “New proteins” dataset in detail in the main text and cited the corresponding figures and tables in the supplementary material.

Now, due to the importance of the “New proteins” dataset, we have splitted the figure originally in the supplementary material and placed it in a subfigure in the main text (Fig. 2d, Extended Data Fig. A1d, Supplementary Fig. 1d, Extended Data Table A3). In addition, in the latest version, we have uniformly introduced the datasets we used in Extended Data Table A8, and added “New protein search test” Section for detailed description.

Please refer to my one-by-one response later for more detailed instructions.

Q 2.2 Major:

1. In the previous review round, I raised a critical concern regarding PLMsearch’s inability to generate sequence-sequence or sequence-structure alignments. This capability is often expected by users, and simply stating that sequences are from the same fold without providing alignment information is not meaningful. I agree with the authors that generating alignments for all query-target pairs in the entire database is expensive and not very useful. Therefore, generating alignments only for the top-ranked candidates is sufficient. However, the method used by the authors to generate alignments, specifically the use of the Smith-Waterman alignment, is not ideal. Smith-Waterman alignment is typically effective for homologous pairs with a sequence identity (SID) greater than 70%, but it is not suitable for remote homologous pairs with SID less than 30%. Taking this approach might risk losing the advantages of PLMsearch. Even traditional profile-based methods such as PSI-BLAST or profile-HMM-based approaches like HHblits and HHsearch are more sensitive in both homologous and remote homologous alignments.

To address this issue, the authors could consider a more suitable approach, i.e., what PLM-BLAST did in their work, which integrates residue-level embeddings from ESM with residue-level profiles from either HHblits or PSI-BLAST profiles and applies these to dynamic programming-based alignment methods, resulting in a more robust alignment. As indicated by your benchmark results, PLM-BLAST demonstrates strong and robust performance. Note that simply relying on PLM-BLAST without any additional implementations may not be appreciated in the next round of revisions, as it may give the impression that your pipeline only consists of a combination of third-party tools.

Reply: Thanks for the comments. As explained in Supplementary Table 5, as a “Search method”, PLM-Search cannot provide any alignment suggestions. To this end, we developed an independent alignment method PLMAlign. The above section “Develop PLMAlign for alignments and alignment scores” details the PLMAlign pipeline. Simply put, PLMAlign can achieve local and global alignment. The pipeline is similar to the SW and NW algorithms. However, by converting a fixed substitution matrix (used in SW and NW) into similarity calculated by the dot product of per-residue embeddings, PLMAlign is able to

capture deep evolutionary information and perform better on remote homology protein pairs. In addition, by using the linear gap penalty, PLMAlign can better align remote homology pairs while reducing the algorithm complexity to $O(mn)$ to ensure high efficiency.

To facilitate comparison, we summarize the main differences between PLMAlign and related methods (SW/NW and pLM-BLAST) in Supplementary Table 7. In addition, we compared the alignment accuracy of different alignment methods on Malisam [4] and Malidup [5] in “Evaluation on remote homology alignment” Supplement Section.

Furthermore, as mentioned above, “Generating alignments for all query-target pairs in the entire database is expensive and not very useful. Therefore, generating alignments only for the top-ranked candidates is sufficient”. To this end, we developed PLMSearch + PLMAlign and emphasized its advantages in performance and efficiency in the main text. We have modified the description in “Ablation experiments: PfamClan, SS-predictor, and PLMAlign makes PLMSearch more robust” and “Discussion” Section.

Q 2.3 2. *The new benchmark results comparing PLM-BLAST, HHblits, and PLMsearch (Extended Data Table A1 and Supplementary Table 2) indicate that PLMsearch does not outperform PLM-BLAST. Specifically, for “new proteins,” PLMsearch performs significantly worse than PLM-BLAST (P@1 values of 0.813 and 0.943, respectively). This suggests that PLMsearch may have a disadvantage in detecting homologous proteins with new or challenging folds and may be overfitting to known or common folds.*

Reply: Thanks for the comments. As we discussed above, PLMSearch as a search method uses “Fast retrieval based on similarity prediction between embeddings”. Correspondingly, as an “Alignment method”, pLM-BLAST uses “Pairwise alignment based on SW/NW algorithm, obtaining similarity through alignment scores”. This enables pLM-BLAST to use per-residue embedding as input, which contains more detailed residue-level information, while the per-protein embedding used by PLMSearch does not. In our latest benchmark (Extended Data Table A1, Extended Data Table A3, briefly summarized in Table R1 and Table R2), it is more appropriate to use PLMAlign and pLM-BLAST for horizontal comparison, because they both conduct pairwise alignments and calculate alignment scores based on per-residue embedding.

In our previous **and present** reported results (Extended Data Table A3, briefly summarized in Table R2), the performance of SS-predictor is always lower than that of pLM-BLAST, both in “SCOPE40-test” and “New proteins”. However, PLMSearch uses PfamClan to pre-filter and achieves higher performance than pLM-BLAST on “SCOPE40-test”. Nevertheless, PfamClan invalids completely on the special dataset of “New proteins”, which causes PLMSearch to degenerate into SS-predictor. Therefore, PLMSearch has the same performance disadvantage as SS-predictor on “New proteins”.

The performance gap has nothing to do with overfitting with known data. Because both “SCOPE40-test” and “New proteins” are unknown data. “New proteins” is only a subset of “SCOPE40-test” (110 among 2,207). “New proteins” contains 110 proteins that failed to scan any Pfam domain in the “SCOPE40-test” as queries. Their main difference is only in the loss of Pfam domain information, which makes PfamClan completely invalid, causing the performance of PLMSearch to degrade exactly like SS-predictor.

Considering the scenario of searching extremely large datasets like UniRef50, SS-predictor and PLMSearch will still only use per-protein embedding for search to provide the best tradeoff between accuracy and speed. If more refined sorting result is needed, PLMAlign is provided to use alignment scores to rerank the search results of SS-predictor and PLMSearch (as what SS-predictor + PLMAlign and PLMSearch + PLMAlign do). Specifically, PLMSearch + PLMAlign only uses PLMAlign to align the protein pairs filtered by PLMSearch with a similarity higher than 0.3 and reranks them. Compared to PLMAlign, PLMSearch + PLMAlign reduces the alignment time (nearly 16 times) while maintaining comparable precision, underscoring the benefits of leveraging PLMSearch to pre-filter (Extended Data Fig. A1b, Extended Data Table A1, briefly summarized in Table R1). In this revision, we have modified the description in “Ablation experiments: PfamClan, SS-predictor, and PLMAlign makes PLMSearch more robust” Section.

Methods	AUROC			AUPR			MAP	P@K		Time
	Fam	Sfam	Fold	Fam	Sfam	Fold	MAP	P@1	P@10	Seconds
Baselines										
pLM-BLAST	0.940	0.642	0.176	0.973	0.779	0.305	0.659	0.921	0.760	18,812 s
Foldseek	0.883	0.584	0.214	0.921	0.703	0.320	0.598	0.908	0.751	12 s
Foldseek-TM	0.898	0.664	0.296	0.906	0.695	0.337	0.626	0.905	0.756	173 s
TM-align	0.935	0.721	0.346	0.971	0.866	0.569	0.781	0.941	0.806	11,303 s
Our methods										
SS-predictor	0.869	0.623	0.225	0.891	0.713	0.324	0.601	0.821	0.686	10 s
PLMSearch	0.928	0.826	0.438	0.931	0.849	0.473	0.685	0.922	0.765	4 s
PLMAlign	0.946	0.652	0.196	0.974	0.807	0.354	0.670	0.919	0.763	12,470 s
SS-predictor + PLMAlign	0.949	0.665	0.211	0.975	0.822	0.391	0.677	0.915	0.763	3,596 s
PLMSearch + PLMAlign	0.933	0.787	0.342	0.956	0.887	0.521	0.660	0.928	0.763	807 s

Table R1 All-versus-all search test on the SCOPE40-test dataset. The definition of AUROC, AUPR, MAP, and P@K is detailed in “Metrics” Section. The highest value achieved for each metric is highlighted in bold. Due to the width limit, Family and Superfamily are abbreviated as Fam and Sfam in the table, respectively. The total search time spent for the all-versus-all search test is recorded.

Methods	MAP	P@K		Time
	MAP	P@1	P@10	Seconds
Baselines				
pLM-BLAST	0.682	0.936	0.805	937.6 s
Foldseek	0.521	0.863	0.730	0.6 s
Foldseek-TM	0.560	0.881	0.740	8.6 s
TM-align	0.776	0.945	0.826	563.3 s
Our methods				
SS-predictor	0.612	0.845	0.712	0.5 s
PLMSearch	0.612	0.845	0.712	0.5 s
PLMAlign	0.692	0.936	0.807	621.5 s
SS-predictor + PLMAlign	0.679	0.927	0.801	179.2 s
PLMSearch + PLMAlign	0.679	0.927	0.801	179.2 s

Table R2 Evaluation on new proteins. See “New protein search test” Section. The definition of MAP, P@K is detailed in “Metrics” Section. The highest value achieved for each metric is highlighted in bold. The total search time spent for the search test is recorded.

Q 2.4 The authors appear to selectively place these disadvantageous data in supplementary materials and avoid discussing performance differences between PLMsearch and PLM-BLAST in the main text, which may not effectively address the issues with PLMsearch.

Reply: Thanks for the comments. In the ablation experiments of the original main text, we had discussed the disadvantages of PLMSearch on the “New proteins” dataset without PfamClan as a component.

Now, due to the importance of the “New proteins” dataset, we have splitted the figure originally in the supplementary material and placed it in a subfigure in the main text (Fig. 2d, Extended Data Fig. A1d, Supplementary Fig. 1d, Extended Data Table A3). In addition, in the latest version, we have uniformly introduced the datasets we used in Extended Data Table A8, and added “New protein search test” Section for detailed description.

Q 2.5 The authors attribute the poor performance of PLMsearch on “new proteins” to the PfamClan method, which relies on PfamScan, a third-party method. However, since your pipeline integrates PfamClan as one component, users are likely to be concerned about the overall performance of your pipeline and may not differentiate between PfamClan and the SS-predictor. If the authors believe that PfamClan is not suitable for detecting remote homologous “new proteins,” they could consider either replacing it with their own method or re-tuning PfamClan.

Reply: Thanks for the comments. For general protein datasets, the probability of being unable to scan any pfam domain is relatively low (below 5%, 4.9% on SCOPe-test (110/2,207), 3.3% on SwissProt (14,204/430,140)). When we test on SCOPe-test and Swiss-Prot, both query and target proteins contain a certain proportion of “New proteins”, and the PLMSearch pipeline will use SS-predictor to return their search results at this time. Although performance is reduced due to the absence of the PfamClan component, it has less impact on the overall performance of the PLMSearch pipeline.

In addition, if the user is only interested in these special “New proteins”, PLMSearch + PLMAlign is recommended. PLMSearch + PLMAlign reranks the search results of PLMSearch using the alignment scores from PLMAlign, and improves precision significantly on “New proteins”. We have modified the description in “PLMsearch reaches similar sensitivities as structure search methods” Section.

For user’s convenience, the PLMSearch web server will now automatically use PLMAlign to align the Top-10 proteins searched by the user, and reranks them using the alignment score (Fig. R1a-b). The choice of Top-10 is due to the limitation of web server computing ability and in order not to affect the search efficiency. If users have additional alignment needs, the PLMAlign web server and source code have been released to support (Fig. R1c-d).

Q 2.6 From a user’s perspective, they may prefer to use PLM-BLAST over PLMsearch since PLM-BLAST is comparable or slightly better than PLMsearch for general fold proteins and significantly better for “new proteins.” Additionally, even for general folds (Extended Data Table A1), the SS-predictor (developed by the authors) performs significantly worse than PLM-BLAST. After integrating the third-party method, PfamScan, into the meta-method PLMsearch, it only achieves comparable results with PLM-BLAST.

Reply: Thanks for the comments. As we discussed above, as a “Search method”, PLMSearch/SS-predictor uses “Fast retrieval based on similarity prediction between embeddings”. Correspondingly, as an “Alignment method”, pLM-BLAST uses “Pairwise alignment based on SW/NW algorithm, obtaining similarity through alignment scores”. This enables pLM-BLAST to use per-residue embedding as input, which contains more detailed residue-level information, while the per-protein embedding used by PLMSearch/SS-predictor does not. In our latest benchmark (Extended Data Table A1, Extended Data Table A3, briefly summarized in Table R1 and Table R2), it is more appropriate to use PLMAlign and pLM -BLAST for horizontal comparison, because they both conduct pairwise alignments and calculate alignment scores based on per-residue embedding.

Overall, PLMSearch, especially our latest PLMSearch + PLMAlign, is able to maintain extremely fast search speeds while being able to search extremely large datasets like UniRef50, and at the same time achieve comparable results with pLM-BLAST (both in “General” and “New” proteins). And as the reviewer mentioned before, enerating alignments for all query-target pairs in the entire database is expensive and not very useful. This is the problem that PLMSearch + PLMAlign mainly solves, which only provides alignments for potential homologous protein pairs filtered by PLMSearch.

Q 2.7 Focusing on limitations in the design of the PLM-BLAST method and ignoring its robust and strong performance is not an effective way to make a comparison and highlight the authors’ method. Moreover, some of the limitations mentioned by the authors regarding PLM-BLAST may actually be considered advantages by some users. For example, PLM-BLAST can generate alignments using residue-level embeddings from a protein language model.

Reply: Thanks for the comments. The primary limitation of pLM-BLAST lies in the maximum size of the target dataset. This is exactly what PLMSearch + PLMAlign solves. Now, when searching large-scale datasets, PLMSearch + PLMAlign can first filter out the large number of low-quality alignments through PLMSearch and obtain alignment results using PLMAlign only for potential homologous protein pairs. In addition, re-ranking search results using alignment scores can provide a more refined search result.

These have been implemented through our web servers and source code (for PLMSearch and PLMAlign). Users can freely choose what they want and obtain it easily.

Q 2.8 *The current PLMsearch pipeline comprises three main components:*

(i) PfamClan, which is primarily based on PfamScan, a third-party method for quickly scanning coordinates for remote homologous proteins. This component plays a major role in saving time during the search.

(ii) The SS-predictor, which is the core component developed by the authors. It predicts similarity scores between the query and coordinates from (i) and subsequently re-ranks them.

(iii) Generation of alignments using the Smith-Waterman algorithm based on BLOSUM62 for sequence-sequence pairs, which is also a third-party method (a module from Biopython).

As mentioned, (i) presents a critical issue due to PfamClan's weakness in detecting remote homologous "new proteins" with new and challenging folds, causing PLMsearch to perform much worse than sequence-based methods like PLM-BLAST and HHblits. Additionally, (iii) employing the classical Smith-Waterman algorithm is not well-suited for remote homologous alignment, which contradicts the main objective of developing a pipeline for remote homologous protein detection.

Regarding the core component (ii) developed by the authors, it does not exhibit significant differences from some older machine learning-based fold recognition methods, except for the integration of the protein language model. Examples from the following papers have quite similar ideas to SS-predictor:

*- Pietro Di Lena, Pierre Baldi, "Fold recognition by scoring protein maps using the congruence coefficient," *Bioinformatics*, Volume 37, Issue 4, February 2021, Pages 506–513.*

*- Bin Liu, Chen-Chen Li, Ke Yan, "DeepSVM-fold: protein fold recognition by combining support vector machines and pairwise sequence similarity scores generated by deep learning networks," *Briefings in Bioinformatics*, Volume 21, Issue 5, September 2020, Pages 1733–1741.*

I would highly recommend that the authors focus on making novel contributions to components (i) and (iii), aiming to strengthen the entire pipeline's performance for both general and challenging fold (new protein) recognition.

Reply: Thanks for the comments. As we mentioned above, we have optimized each of the three main components, which I briefly summarize here:

(1) For PfamClan: As we described in "Pfam Database Update" above, we have recently updated the Pfam dataset to its latest version (Pfam36.0, 2023-09-12) and reused PfamScan to scan the protein domains, which further reduces the number of "New proteins" (from 123 to 110) and improves the performance of PLMSearch (about 1% overall). As deep learning methods such as ProtENN [6] are used in the expansion of the Pfam dataset, we believe that more and more pfam domains will be scanned, and the ratio of "New proteins" will reduce. Currently, the ratio of "New proteins" is roughly below 5% (4.9% on SCOPe-test (110/2,207), 3.3% on SwissProt (14,204/430,140)). Therefore, we are committed to regularly updating Pfam dataset to ensure PLMSearch's ongoing accuracy and relevance.

(2) For SS-predictor: As we described in "Add CATHS40 as training data and summarize all datasets" above, in order to further improve the generality and performance of SS-predictor, we have newly added CATHS40 as training data. CATHS40 further expands the proteins and protein pairs with TM-score above 0.5 in our training data, which slightly improved the performance of SS-predictor, leading to an improvement in the performance of PLMSearch as well (about 3% overall, in both "General" and "New" proteins).

(3) For alignments: As we described in "Develop PLMAlign for alignments and alignment scores", we have developed PLMAlign to align the searched protein pairs and calculate the alignment scores. The performance of directly using PLMAlign to align all protein pairs and rank with alignment scores slightly exceeds that of pLM-BLAST. Moreover, PLMSearch + PLMAlign only uses PLMAlign to align the protein pairs filtered by PLMSearch with a similarity higher than 0.3 and reranks them. Compared to PLMAlign, PLMSearch + PLMAlign reduces the alignment time (nearly 16 times) while maintaining

comparable precision, underscoring the benefits of leveraging PLMSearch to pre-filter (Extended Data Fig. A1b, Extended Data Table A1, briefly summarized in Table R1).

Through the above improvements, we have improved the performance and versatility of PLMSearch for both general and challenging fold (“New proteins”) recognition. In addition, as the latest contribution, PLMSearch + PLMAlign can first filter out the vast majority of low-quality protein pairs and generate alignments for the retained potential homologous protein pairs.

With efficient and user-friendly web server and source code (Fig. R1), we believe PLMSearch and PLMAlign are ready to serve researchers now.

Reviewer 3

Q 3.1 I believe that the authors have mostly addressed my comments and the manuscript is ready for publication.

Reply: Thank you for your valuable comments and suggestions which have improved our manuscript.

References

- [1] Sillitoe, I., Bordin, N., Dawson, N., Waman, V.P., Ashford, P., Scholes, H.M., Pang, C.S., Woodridge, L., Rauer, C., Sen, N., *et al.*: Cath: increased structural coverage of functional space. *Nucleic acids research* **49**(D1), 266–273 (2021)
- [2] Hamamsy, T., Morton, J.T., Blackwell, R., Berenberg, D., Carriero, N., Gligorijevic, V., Strauss, C.E., Leman, J.K., Cho, K., Bonneau, R.: Protein remote homology detection and structural alignment using deep learning. *Nature biotechnology*, 1–11 (2023)
- [3] Kaminski, K., Ludwiczak, J., Pawlicki, K., Alva, V., Dunin-Horkawicz, S.: plm-blast – distant homology detection based on direct comparison of sequence representations from protein language models. *bioRxiv* (2023)
- [4] Cheng, H., Kim, B.-H., Grishin, N.V.: Malisam: a database of structurally analogous motifs in proteins. *Nucleic acids research* **36**(suppl_1), 211–217 (2007)
- [5] Van Heel, A.J., de Jong, A., Montalban-Lopez, M., Kok, J., Kuipers, O.P.: Bagel3: automated identification of genes encoding bacteriocins and (non-) bactericidal posttranslationally modified peptides. *Nucleic acids research* **41**(W1), 448–453 (2013)
- [6] Bileschi, M.L., Belanger, D., Bryant, D.H., Sanderson, T., Carter, B., Sculley, D., Bateman, A., DePristo, M.A., Colwell, L.J.: Using deep learning to annotate the protein universe. *Nature Biotechnology* **40**(6), 932–937 (2022)

Reviewer #1 (Remarks to the Author):

The authors have addressed all my questions.

The new evidence suggests that the competing software pLM-BLAST outperforms the presented software in several benchmarks while the winning benchmarks present only marginal gains (and in some cases only when combined with other software while software combinations were not benchmarked with competing software).

In addition, the advertised feature of full protein embeddings will be rather useful for a niche audience and not for a broad life science audience.

While my technical concerns were now fully addressed, I do question the impact the presented software will have for a broad audience. I personally see this software placed in a more specialised journal, but leave it up to the editors to decide how much value is added by this study.

Reviewer #2 (Remarks to the Author):

I greatly appreciate the authors' dedication and patience in thoroughly addressing all my concerns. I am particularly pleased to note their awareness of the issues in sequence alignment and their efforts in developing PLMAlign to resolve these questions. The integration of PLMSearch with PLMAlign has resulted in a robust and sophisticated pipeline. In my opinion, this now stands as a highly effective tool in the field. All of my concerns have been clearly resolved, leaving me with no further comments. Congratulations on this achievement!

Reviewer #2 (Remarks on code availability):

The code is well-structured and the README file is sufficiently clear.

Response Letter

Dear Reviewers,

We greatly appreciate your time and efforts in reviewing the manuscript. We thank the reviewers for the highly pertinent and helpful comments.

In the following, we present one-by-one responses to the reviewers' comments.

Reviewer 1

Q 1.1 *The authors have addressed all my questions.*

The new evidence suggests that the competing software pLM-BLAST outperforms the presented software in several benchmarks while the winning benchmarks present only marginal gains (and in some cases only when combined with other software while software combinations were not benchmarked with competing software).

In addition, the advertised feature of full protein embeddings will be rather useful for a niche audience and not for a broad life science audience.

While my technical concerns were now fully addressed, I do question the impact the presented software will have for a broad audience. I personally see this software placed in a more specialised journal, but leave it up to the editors to decide how much value is added by this study.

Reply: Thanks for the comments. It is glad to know that we successfully solved all your questions.

Experiments have shown that compared to pLM-BLAST, PLMAlign (the winning benchmark mentioned above) presents only marginal gains. But more importantly, by using PLMSearch for pre-filtering, PLMSearch + PLMAlign can filter out numerous low similarity and meaningless alignments, thereby maintaining high efficiency (approximately 16 times improvement) while ensuring the quality of search and alignment.

In fact, we consider such a rather broad usage scenario. When biological researchers want to search their own protein sequence data against large-scale standard datasets (like UniRef50, with 53.6M proteins), the most commonly used tool is still MMseqs2. Because many of the latest sequence search methods, including pLM-BLAST, are difficult to achieve both search sensitivity and efficiency in such a scenario. This is the main problem that PLMSearch wants to solve, and the use of full protein embeddings is mainly to solve the memory challenges and speed challenges faced when searching these large-scale datasets.

Reviewer 2

Q 2.1 I greatly appreciate the authors' dedication and patience in thoroughly addressing all my concerns. I am particularly pleased to note their awareness of the issues in sequence alignment and their efforts in developing PLMAlign to resolve these questions. The integration of PLMSearch with PLMAlign has resulted in a robust and sophisticated pipeline. In my opinion, this now stands as a highly effective tool in the field. All of my concerns have been clearly resolved, leaving me with no further comments. Congratulations on this achievement!

Reply: Thank you for your valuable comments and suggestions which have improved our manuscript.

Q 2.2 The code is well-structured and the README file is sufficiently clear.

Reply: Thanks for the encouraging comments. Improving the readability and reproducibility of code has always been one of our goals. We really hope everyone enjoys PLMSearch.